# FrameShield: Adversarially Robust Video Anomaly Detection

**Mojtaba Nafez**
Department of Computer Engineering
Sharif University of Technology
mojtaba.nafez.99@gmail.com

**Mobina Poulaei** *
Department of Computer Engineering
Sharif University of Technology
m.poulaei@gmail.com

**Nikan Vasei** *
Department of Computer Engineering
Sharif University of Technology
nikanvsiuni@gmail.com

**Bardia Soltani Moakhar**
Department of Industrial Engineering
Sharif University of Technology
bardisoltan@gmail.com

**Mohammad Sabokrou**
Machine Learning and Data Science Unit
Okinawa Institute of Science and Technology
mohammad.sabokrou@oist.jp

**MohammadHossein Rohban**
Department of Computer Engineering
Sharif University of Technology
rohban@sharif.edu

## Abstract

Weakly Supervised Video Anomaly Detection (WSVAD) has achieved notable advancements, yet existing models remain vulnerable to adversarial attacks, limiting their reliability. Due to the inherent constraints of weak supervision—where only video-level labels are provided despite the need for frame-level predictions—traditional adversarial defense mechanisms, such as adversarial training, are not effective since video-level adversarial perturbations are typically weak and inadequate. To address this limitation, pseudo-labels generated directly from the model can enable frame-level adversarial training; however, these pseudo-labels are inherently noisy, significantly degrading performance. We therefore introduce a novel Pseudo-Anomaly Generation method called Spatiotemporal Region Distortion (SRD), which creates synthetic anomalies by applying severe augmentations to localized regions in normal videos while preserving temporal consistency. Integrating these precisely annotated synthetic anomalies with the noisy pseudo-labels substantially reduces label noise, enabling effective adversarial training. Extensive experiments demonstrate that our method significantly enhances the robustness of WSVAD models against adversarial attacks, outperforming state-of-the-art methods by an average of 71.0% in overall AUROC performance across multiple benchmarks. The implementation and code are publicly available at https://github.com/rohban-lab/FrameShield.

## 1 Introduction

Video Anomaly Detection (VAD) is a fundamental component of surveillance systems, with applications spanning public safety, healthcare, and industrial monitoring, identifying rare and hazardous events such as accidents, violence, and equipment malfunctions Gopalakrishnan [2012], Sultani et al. [2018]. In recent years, due to the labor-intensive nature of frame-level labeling, research

---

*Equal Contribution

39th Conference on Neural Information Processing Systems (NeurIPS 2025).

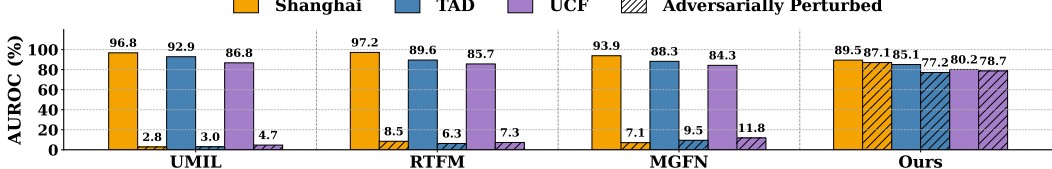

Figure 1: **Robustness Assessment of Video Anomaly Detection (VAD) Methods**: A comparative evaluation of SOTA VAD methods on well-established benchmarks: Shanghai, TAD, and UCF Crime, under both standard conditions and adversarial attack scenarios. The results highlight the vulnerability of existing SOTA methods and demonstrate the superior robustness and reliability of our proposed method, FrameShield, in both clean and adversarial settings.

has shifted towards Weakly Supervised Video Anomaly Detection (WSVAD) Majhi et al. [2021] Yang et al. [2022] Jia et al. [2023] . Ensuring robustness against adversarial attacks is crucial for deploying machine learning models in critical and high-reliability scenarios Rony et al. [2019]. These attacks introduce subtle, almost imperceptible perturbations into the input video, causing models to misclassify normal frames as anomalies and vice versa. Although SOTA VAD methods have demonstrated near-optimal performance under standard conditions, their susceptibility to adversarial perturbations results in substantial performance degradation, as illustrated in Figure 1, raising serious concerns about their reliability and robustness in real-world applications.

Despite advances in VAD, WSVAD's adversarial robustness remains largely unexplored. Enhancing its robustness presents significant challenges. First, current WSVAD methods rely heavily on pretrained feature extractors like I3D Carreira and Zisserman [2017], C3D Tran et al. [2015], Swin-Transformer Liu et al. [2021], and CLIP Radford et al. [2021], which, despite strong representational power, are highly susceptible to adversarial attacks. Second, adversarial training (AT)—a widely used defense mechanism for improving model robustness through the augmentation of training data with adversarial examples—faces unique challenges in WSVAD. This is mainly due to the inherent constraints of the Multiple Instance Learning (MIL) framework, where only video-level labels are available during training, while frame-level predictions are required during inference Jafarinia et al. [2024].

In WSVAD, MIL-based loss functions are commonly employed, where an aggregator function such as max pooling is applied to frame-level outputs to produce a video-level prediction that matches the available label, enabling cross-entropy loss for training Sultani et al. [2018]. During adversarial training, perturbations are applied to the entire video. However, only the features selected by the aggregator such as the maximum-valued feature are adversarially influenced, since the gradient primarily flows through that specific component. This design introduces a critical vulnerability. During training, perturbations are applied only to the features chosen by the aggregator (e.g., the maximum). In contrast, during inference, attackers are not constrained in this way and can manipulate entire frames, producing more localized and impactful perturbations across all features. The absence of frame-level annotations in training constrains adversarial example generation to video-level supervision, leading to weaker perturbations that reduce the robustness of WSVAD models against attacks Mirzaei et al. [2024a], Chen et al. [2021]. We provide a theoretical analysis of this phenomenon in Section 4, demonstrating how max-based aggregation neglects perturbations on non-maximal frames, leaving them exposed to attacks during inference.

To address these limitations, we propose FrameShield, a novel end-to-end adversarial training pipeline designed to address the limitations of pretrained models by fine-tuning the feature extractor. The training process is structured into two main phases. First, we perform standard training using a simple yet effective WSVAD method, generating predicted labels that serve as pseudo-labels. In the second phase, we employ these frame-level pseudo-labels to perform adversarial training, crafting stronger adversarial examples to enhance model robustness. However, as shown in Table 2, the localization performance of SOTA WSVAD methods on the anomaly segments of benchmarks remains suboptimal, often barely surpassing random detection. Our method similarly struggles with anomaly localization, producing noisy pseudo-labels that lead to false positives and false negatives, particularly in anomalous videos. In contrast, all frames of normal videos are consistently labeled

without errors. As a result, the presence of noisy pseudo-labels leaves our method vulnerable to adversarial attacks on anomaly videos Dong et al. [2023].

To address this vulnerability, we present Spatiotemporal Region Distortion (SRD), an innovative Pseudo-Anomaly Video Generation method designed to produce synthetic anomalies with accurate frame-level annotations. SRD works by randomly selecting an intermediate frame from a normal video, utilizing Grad-CAM Selvaraju et al. [2017] to identify the foreground objects, and then performing multiple harsh augmentations on the largest connected component within that region. To simulate the appearance of moving anomalies throughout the video sequence, we introduce motion irregularities Zhu and Newsam [2019], Yang et al. [2021] by defining a randomly curved vector, directing the corrupted region's displacement across consecutive frames with additional harsh augmentations. This technique enables the generation of synthetic anomaly videos with precise frame-level labeling, eliminating the need for extra supervision and enhancing adversarial training. By merging these accurately labeled synthetic anomalies with insights into real anomaly distributions derived from pseudo-labels, we introduce our adversarially robust WSVAD framework, FrameShield.

**Contribution** We introduce FrameShield, the first adversarial training pipeline specifically designed to enhance the robustness of WSVAD models against adversarial attacks. Our method employs frame-level adversarial training within a weakly supervised setup, leveraging real anomaly distributions from pseudo-labels and mitigating false positives and negatives error through Spatiotemporal Region Distortion (SRD) for precise frame-level annotations. We theoretically justify the superiority of supervised adversarial training over MIL-based approaches. FrameShield is evaluated against strong attack methods, including PGD-1000 Madry et al. [2017], AA Croce and Hein [2020], and $A^3$ Liu et al. [2022]. Experimental results, on average across well-established benchmarks, demonstrate a near 53% improvement in overall AUROC for robust performance and 68.5% in anomaly segments, while maintaining competitive performance on standard setup.

## 2 Related Work

VAD is vital in surveillance, public safety, and automated monitoring. Traditional fully-supervised methods demand costly frame-level annotations due to the rarity of anomalies. WSVAD addresses this by using only video-level labels, leveraging Multiple Instance Learning (MIL) to treat each video as a bag of frames, assuming anomalies exist in positive samples. Early MIL-based VAD methods faced challenges with noisy labels and weak temporal modeling Sultani et al. [2018]. Improvements followed with noise suppression, temporal modeling (e.g., MGFN Chen et al. [2022]), dual memory units (UR-DMU Zhou et al. [2023]), and unbiased training (UMIL Lv et al. [2023]) through feature clustering and contrastive loss. Recently, Vision-Language Models (VLMs) like CLIP enhanced anomaly detection by capturing visual and semantic cues Joo et al. [2023], Wu et al. [2024], Chen et al. [2023]. However, WSVAD models remain vulnerable to adversarial attacks, as they rely on non-robust pre-trained backbones (e.g., I3D, C3D, Swin Transformer, CLIP) Schlarmann et al. [2024], Chen et al. [2019], Li et al. [2021]. The lack of frame-level annotations further complicates adversarial defense, exposing models to real-world threats. For further information and detailed discussion of the related works, please refer to Appendix M.

## 3 Preliminaries

**Weakly Supervised Video Anomaly Detection (WSVAD)**: Video Anomaly Detection (VAD) is the task of identifying unusual or abnormal events within a video and determining their temporal locations at the frame level. In the WSVAD setup, only video-level supervision is available during training, indicating whether a video contains anomalies, without providing specific frame-level labels. During inference, a model $F_\Theta$ processes a video $V$ containing $N$ frames and generates an anomaly score $S_i(F_\Theta; V)$ for each frame $i$. If the score of a frame surpasses a predefined threshold, that frame is classified as anomalous; otherwise, it is considered normal.

**Adversarial Attack on Video Anomaly Detectors**: Adversarial attacks, commonly studied in the context of classification tasks, involve intentionally modifying an input sample $x$ with its corresponding label $y$ to generate a new sample $x^*$ that increases the model's prediction error by maximizing the loss function $\ell(x^*; y)$ Yuan et al. [2019], Xu et al. [2019]. The resulting input $x^*$ is referred to as an *adversarial example*, and the difference $x^* - x$ is called the *adversarial perturbation*. To ensure

that the adversarial example remains semantically similar to the original input, the perturbation is constrained such that its $l_p$-norm does not exceed a predefined threshold $\epsilon$. Formally, an adversarial example satisfies the condition $x^* = \arg\max_{x':\|x-x'\|_p \leq \epsilon} \ell(x'; y)$ One of the most commonly used and effective techniques for crafting adversarial examples is the *Projected Gradient Descent (PGD)* method Madry et al. [2017], which iteratively updates the input in the direction of the gradient sign of $\ell(x^*; y)$ using a step size $\alpha$.

In this work, we adapt the adversarial attack paradigm to the domain of Video Anomaly Detection (VAD), introducing a targeted, task-specific attack that manipulates videos based on the anomaly scores of individual frames, rather than optimizing against an overall loss function—most existing methods in WSVAD rely on MIL-based losses. Our goal is to mislead the model by **increasing the anomaly scores of normal frames** and **decreasing those of abnormal frames**, which we experimentally demonstrate to be a more effective form of attack (Table 14). The attack is formulated as follows. Starting from the original video $V_0^* = V$, we iteratively update the adversarial video using the rule:

$$V_{t+1}^* = V_t^* + Y \cdot \alpha \cdot \text{sign}\left(\nabla_V S(F_\Theta; V_t^*)\right),$$

where $S(F_\Theta; V_t^*)$ denotes the anomaly scores predicted by the model $F_\Theta$ for each frame of the video $V_t^*$, and $\alpha$ is the step size. The vector $Y$ is defined such that $Y_i = +1$ for normal frames and $Y_i = -1$ for anomalous frames, with $i$ indexing the frame position.

## 4 Methods

**Theoretical Motivation.** We hypothesize that using $\max$ as the MIL aggregator results in weak attacks on instances, or frames. Note that by denoting $x = (x_1, \ldots, x_k)$ as $k$ video frames, the gradient of loss with respect to the input $x$ becomes:

$$\nabla_x l(\max(f(x_1), \ldots, f(x_k)), y) = l'(f, y) . \nabla_x f(x_j),$$

where $j$ is the index of the frame leading to the maximum, i.e. $j = argmax_i f(x_i)$ for the specific input $x$. This results in the gradient-based attack to be applied only on a *single* frame. Once such attack is used for training, the base classifier $f$ would become robust with respect to a subset of frames, i.e. only those with maximum score. On the other hand, there could be other frames $j' \neq j$, where $f(x_{j'})$ is also high, though be a bit smaller than $f(x_j)$. The attack does not consider such frames, and if $x_{j'}$ does not follow the same distribution as $x_j$, the model adversarially trained base classifier based on this attack would fail to generalize robustness on $x_{j'}$. One could use other soft versions of the $\max$, such as Log-Sum-Exp (LSE) to mitigate this issue. However, our experiments in Table 3 indicate that although LSE outperforms the max function, it remains ineffective in reducing the performance of clean-trained models to zero under adversarial attacks. We note that such operators decrease model sensitivity to a single/small number of frames.

For these reasons, we prefer training attacks that are applied on every single frame but does not alter the $\max$ operator to not compromise the model sensitivity to outliers. This could be achieved by directly attacking $f(x_i)$ for all $i$. Here, the attack is designed based on:

$$L := \max_{\|\delta_i\|_\infty \leq \epsilon} l(f(x_i + \delta_i), f(x_i)), \tag{1}$$

i.e. make the model to not change its original prediction for *every* frame in a given input video. Here, we consider $f(x_i)$ as a pseudo-label and design the attack based on it. The loss in Eq. 1 closely resembles what is known as the "boundary error," as opposed to "natural error" in the so-called TRADES method Zhang et al. [2019a]. Such attacks could serve as a *regularization for robustness*. Here, one could aim for optimizing the standard error added by the adversarial loss in Eq. 1 to achieve a better trade-off between the standard and adversarial errors. This loss is indeed was shown to be an almost sharp upper-bound on the difference between the robust risk and *optimal standard risk* Zhang et al. [2019a]:

$$\mathcal{R}_{rob}(f) - \mathcal{R}_{nat}^\star \leq \psi^{-1}(\mathcal{R}_l(f) - \mathcal{R}_l^\star) + \mathbb{E}(L),$$

where $\mathcal{R}_{rob}$ and $\mathcal{R}_{nat}^\star$ represent the robust and optimal standard risks, respectively. Furthermore, $\psi$ is a non-decreasing function, and $\mathcal{R}_l$ represents the risk with respect to the loss function $l$, and $L$ is defined in Eq. 1. Therefore, this loss could be an excellent alternative in weakly supervised scenarios where the ground truth labels are missing for many instances.

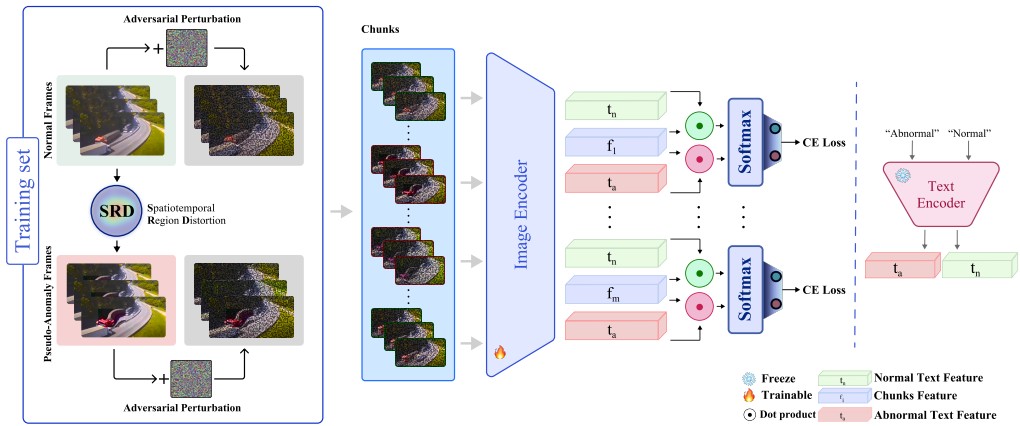

Figure 2: Overview of the FrameShield Framework: (1) The WSVAD training set is constructed using frame-level labels for normal data and frame-level pseudo labels for weakly labeled real anomaly data. Additionally, the Spatiotemporal Region Distortion (SRD) module generates pseudo-anomaly samples with precise frame-level annotations, further augmenting the training set with adversarial perturbations. (2) Two predefined prompts, "normal" and "anomaly," are used to extract text embeddings from the frozen text encoder. (3) Training videos are segmented into chunks and processed by the XClip-based encoder. The dot product between each chunk's feature representation and the text prompt embeddings is computed to obtain normality and abnormality scores, optimizing the network through chunk-level cross-entropy loss.

**Overview.** Current WSVAD methods exhibit significant vulnerability to adversarial attacks. Our experiments and analysis of the MIL-based loss function in Table 3 and Section 6 underscores the necessity of frame-level labeling to enhance adversarial robustness. To address this challenge, we introduce FrameShield, a novel approach that strengthens model resilience by leveraging weakly labeled real abnormal data through pseudo-label generation and precisely labeled chunk pseudo-anomalies. FrameShield operates in two main stages: first, the WSVAD model undergoes standard training using an MIL-based loss function, allowing it to learn anomaly patterns effectively. This learned knowledge is then utilized to generate pseudo labels for the anomaly subset of the training data. In the second stage, the adapted WSVAD model is adversarially trained with both pseudo labels and pseudo anomalies, providing more granular supervision at the frame level and improving its robustness against adversarial manipulations. The following sections provide a detailed breakdown of each stage in FrameShield's training pipeline.

## 4.1 First Phase: PromptMIL Training

Our proposed Weakly Supervised Video Anomaly Detection (WSVAD) method, **PromptMIL**, partitions each video into $m$ chunks, denoted as $v_i$, where $i \in \{1, 2, \ldots, m\}$. We employ **X-Clip** Ma et al. [2022] as the feature extractor, represented as $F_\Theta$. Each video chunk $v_i$ is processed through $F_\Theta$, generating a corresponding feature vector $\mathbf{f}_i$.

Additionally, we extract feature vectors for two specific text prompts: **"Normal"** and **"Abnormal"**, using the X-Clip text encoder. For each video chunk, the dot product is computed between its feature vector $\mathbf{f}_i$ and the feature vectors of the two text prompts. We then apply a **softmax** function to produce a probability distribution that represents the likelihood of the chunk being normal or abnormal:

$$S_i(F_\Theta; V), (1 - S_i(F_\Theta; V)) = \text{softmax}(\mathbf{f}_i \cdot \mathbf{t}_a, \mathbf{f}_i \cdot \mathbf{t}_n) \tag{2}$$

where $F_\Theta$ represents our FrameShield model, and $\mathbf{t}_n$ and $\mathbf{t}_a$ are the text feature vectors for "Normal" and "Abnormal," respectively. Here, $S_i(F_\Theta; V)$ denotes the predicted anomaly score for the $i$-th chunk. After processing all chunks, we obtain the normality and abnormality probabilities for each chunk. We then aggregate the anomaly scores across all chunks using a Multiple Instance Learning (MIL) **max** aggregator, which selects the maximum anomaly score from the chunks. This

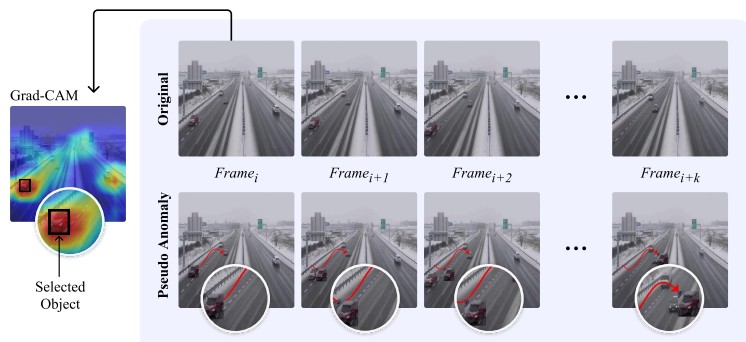

Figure 3: **Visualization of Spatiotemporal Region Distortion (SRD)**: A synthetic anomaly generation method for precise frame-level labeling, applying harsh augmentations to Grad-CAM-identified foreground regions and simulating motion irregularities with random curved vectors across frames.

aggregation step is followed by a **Binary Cross-Entropy loss** calculation, which is directly applied to the maximum anomaly score rather than summing over all chunks:

$$\mathcal{L} = \text{BCE}(\max(S_1(F_\Theta; V), S_2(F_\Theta; V), \dots, S_m(F_\Theta; V)), y) \tag{3}$$

where $m$ is the total number of chunks, and $y$ is the true label of the video. This formulation ensures that the model is optimized based on the highest anomaly score across the chunks, which is particularly effective for identifying the most critical abnormal segments in the video.

**Inference.** During inference, the video is similarly partitioned into $m$ chunks and processed through the feature extractor $F_\Theta$. For each chunk, we perform the dot product operation with the text prompts and apply the softmax to obtain the normality and abnormality scores. Chunks with abnormality score $> \tau = 0.5$ are labeled abnormal; others are normal. Frame-level predictions are obtained by duplicating each chunk's score across its frames. An ablation on $\tau$ is provided in Appendix D.

**Pseudo Label Generation.** Once the PromptMIL model is trained, we utilize it to generate pseudo labels for each chunk of the anomaly videos in the training set. Notably, frames in normal videos are inherently labeled as normal and therefore do not require pseudo labeling. At this stage, our objective is to label the training data—a task that is inherently less challenging, as the model has already been trained on it. Details regarding the alternative methods for pseudo-label generation and the performance evaluation of our PromptMIL model can be found in Appendix B.

## 4.2 Second Phase: Adversarial Training

During this training phase, we conduct adversarial training on a constructed fully supervised VAD model by leveraging precise chunk-level labels for normal videos, alongside generated pseudo-labels for real abnormal chunks. Additionally, we incorporate exact chunk-level annotations for the pseudo-anomalies generated by our Spatiotemporal Region Distortion (SRD) method. This approach is specifically designed to mitigate label noise within the pseudo-labeled data, enhancing the model's robustness and accuracy. The following sections provide a comprehensive breakdown of our pseudo-anomaly generation process, the detailed training procedure, the employed loss functions, and the specifics of the adversarial training strategy.

**Spatiotemporal Region Distortion (SRD).** Based on our observations in Table 5 and the analysis presented in Appendix F, we found that solely relying on adversarial training with our generated pseudo-labels is ineffective due to the presence of false positives and false negatives, which hinder proper optimization. To address this issue, we propose a novel yet straightforward pseudo-anomaly generation method called Spatiotemporal Region Distortion (SRD), designed to provide anomalies with precise frame-level annotations and rectify the errors in pseudo-labeled videos.

We recognize that an effective pseudo-anomaly in the video domain should meet three key criteria. First, there should be a high likelihood that the distorted data appears abnormal. Second, the generated data should be near normal samples' distribution, sharing similar semantic and stylistic attributes. This aligns with existing literature on adversarial robustness, which emphasizes the benefits of

Table 1: Frame-level detection performance (% AUROC) of various Video Anomaly Detection (VAD) methods compared to FrameShield across multiple benchmarks, evaluated under both clean settings and adversarial attacks setup (PGD-1000, $\epsilon = \frac{0.5}{255}$) over the entire test set ($AUC_O$).

| Method | Attack | Dataset | | | | |
|--------|--------|---------|---|---|---|---|
| | | UCSD Ped2 | Shanghai | TAD | UCF Crime | MSAD |
| RTFM | Clean / PGD | 98.6 / 2.4 | 97.21 / 8.5 | 89.6 / 6.3 | 85.7 / 7.3 | 86.6 / 10.0 |
| TEVAD | Clean / PGD | 98.7 / 5.8 | 98.1 / 8.4 | 92.3 / 7.6 | 84.9 / 0.0 | 86.82 / 6.5 |
| MGFN | Clean / PGD | 96.3 / 5.0 | 93.9 / 7.1 | 88.3 / 9.5 | 84.3 / 11.8 | 84.9 / 12.1 |
| Base MIL | Clean / PGD | 92.3 / 7.3 | 95.2 / 0.6 | 89.1 / 0.9 | 80.7 / 4.6 | 80.5 / 1.3 |
| UMIL | Clean / PGD | 94.2 / 6.9 | 96.8 / 2.8 | 92.9 / 3.0 | 86.8 / 4.7 | 83.8 / 6.1 |
| UR-DMU | Clean / PGD | 97.3 / 4.1 | 96.2 / 11.2 | - / - | 86.75 / 6.7 | 86.12 / 10.2 |
| VAD-CLIP | Clean / PGD | 98.4 / 6.3 | 97.5 / 3.6 | 92.7 / 5.1 | 88.0 / 8.2 | - / - |
| **Ours** | Clean / PGD | 97.1 / **81.3** | 89.5 / **87.1** | 85.1 / **77.2** | 80.2 / **78.7** | 78.9 / **76.2** |

Table 2: Frame-level detection performance (% AUROC) of various Video Anomaly Detection (VAD) methods compared to FrameShield across multiple benchmarks, evaluated under clean settings and adversarial attacks setup (PGD-1000, $\epsilon = \frac{0.5}{255}$) on the anomaly sections of the test set ($AUC_A$).

| Method | Attack | Dataset | | | | |
|--------|--------|---------|---|---|---|---|
| | | UCSD Ped2 | Shanghai | TAD | UCF Crime | MSAD |
| RTFM | Clean / PGD | 98.01 / 5.1 | 64.31 / 8.5 | 53.08 / 5.0 | 63.86 / 10.2 | 72.35 / 3.6 |
| TEVAD | Clean / PGD | 98.20 / 7.5 | 67.9 / 10.5 | 60.5 / 7.2 | 60.3 / 2.6 | 71.6 / 7.8 |
| MGFN | Clean / PGD | 96.3 / 4.3 | 66.9 / 7.8 | 51.56 / 11.7 | 64.9 / 13.3 | 74.53 / 9.0 |
| Base MIL | Clean / PGD | 90.4 / 8.6 | 63.5 / 3.3 | 56.5 / 4.2 | 60.6 / 4.7 | 63.5 / 3.2 |
| UMIL | Clean / PGD | 91.3 / 7.2 | 69.1 / 5.9 | 65.8 / 5.8 | 68.7 / 6.2 | 72.2 / 8.3 |
| UR-DMU | Clean / PGD | 93.5 / 5.0 | 65.7 / 10.4 | - / - | 68.82 / 7.2 | 68.4 / 7.3 |
| VAD-CLIP | Clean / PGD | 96.9 / 5.7 | 70.2 / 5.9 | 61.9 / 8.3 | 69.3 / 8.0 | - / - |
| **Ours** | Clean / PGD | 94.3 / **91.3** | 62.3 / **61.9** | 50.9 / **30.0** | 60.1 / **53.4** | 64.4 / **60.2** |

decision boundary samples that are near the distribution for enhancing model robustness Xing et al. [2022]. Finally, it is crucial to incorporate temporal characteristics into the anomalies, reflecting unexpected variations over time, such as abrupt speed shifts, sudden motion disruptions, or irregular event sequences that disrupt the normal flow of activities.

Building on our insights into pseudo-anomaly generation in the video domain, we propose Spatiotemporal Region Distortion (SRD). SRD begins by randomly selecting a continuous sequence of frames from a normal video and extracting the initial frame. To identify object regions, Grad-CAM is applied using a pre-trained ResNet18 He et al. [2016], Nafez et al. [2025] model, effectively highlighting the most salient foreground areas. The resulting saliency map is then thresholded to isolate the most prominent features, after which the largest connected component is computed. A bounding rectangle is fitted around this region (with some randomness introduced for generalization), serving as the foundation for a binary mask. Finally, we apply k harsh augmentations, randomly chosen from a predefined set of N aggressive transformations known to disrupt semantic integrity, as supported by prior research Sinha et al. [2021], DeVries and Taylor [2017], Ghiasi et al. [2021], Zhang et al. [2018], Mirzaei et al. [2024b, 2025]. These augmentations are applied exclusively to the masked region, maximizing the chances of the transformed video being perceived as abnormal. For further details, please refer to Appendix A.

To introduce temporal characteristics into the anomaly, SRD defines a randomly curved vector that originates from the center of the rectangle and extends in a random direction. The masked region is then duplicated, distorted with a set of new augmentations, and positioned in the subsequent frame according to the vector's trajectory. This movement progresses step by step through the frame sequence, with each step covering a distance proportional to the vector's total length divided by the number of frames in the sequence. This synchronized motion effectively simulates spatiotemporal anomaly propagation throughout the video. An illustrative example of SRD applied to a video sequence is presented in Figure 3.

Table 3: Frame-level detection performance (% AUROC) of various aggregation methods under adversarial (PGD-1000, $\epsilon = \frac{2}{255}$) conditions, evaluated only on abnormal test videos ($AUC_A$).

| Aggregator | TAD | UCF Crime | MSAD |
|---|---|---|---|
| Max | 45.4 | 51.2 | 48.6 |
| Log-Sum-Exp | 43.8 | 39.3 | 43.0 |
| SmoothMax | 49.7 | 48.7 | 47.1 |
| ABMIL (Attention) | 43.6 | 45.2 | 44.5 |
| **Frame-level** | 0.4 | 0.0 | 0.6 |

Table 4: Frame-level detection performance (% AUROC) of adversarial attacks ($\epsilon = \frac{0.5}{255}$) on our method across three datasets. $AUC_O$ and $AUC_A$ refer to overall and abnormal-only AUC, respectively.

| Dataset | PGD-1000 | | AutoAttack | | $A^3$ | |
|---|---|---|---|---|---|---|
| | $AUC_O$ | $AUC_A$ | $AUC_O$ | $AUC_A$ | $AUC_O$ | $AUC_A$ |
| TAD | 77.2 | 30.0 | 73.1 | 27.3 | 73.6 | 27.2 |
| UCF Crime | 78.7 | 53.4 | 72.1 | 50.3 | 71.5 | 48.7 |
| MSAD | 76.2 | 60.2 | 74.7 | 58.2 | 73.6 | 56.9 |

**Training Process.** In this phase, we leverage the availability of frame-level annotations for normal videos, real abnormal videos, and the pseudo-anomaly videos generated by SRD. Unlike traditional MIL-based training, which relies on video-level loss functions, we shift to a fully supervised learning paradigm. This enables the model to learn more granular representations by directly optimizing chunk-level predictions. As illustrated in Figure 2, the loss function is computed independently for each chunk, allowing for fine-grained supervision:

$$\mathcal{L}_{chunk-wise}(V, Y) = \sum_{i=1}^{m} \text{BCE}(S_i(F_\Theta; V), y_i) \tag{4}$$

where $m$ is the total number of chunks, and $y_i$ is its corresponding ground truth label. Additionally, $Y$ denotes the complete set of chunk-wise labels. This chunk-wise cross-entropy calculation ensures that the model is updated with finer granularity. Moreover, this supervised strategy allows us to apply **strong adversarial perturbations** to the input video during training, effectively building a more robust VAD model against adversarial attacks.

**Adversarial Training of WSVAD.** Given an input video sample $V$, an adversarial version $V_{\text{adv}}$ is crafted by introducing a perturbation $\delta^*$ generated through the PGD-10 attack. This perturbation is constrained by the $l_\infty$ norm with $\epsilon = \frac{0.5}{255}$ and is optimized according to our chunk-wise loss function:

$$\delta^* = \underset{\|\delta\|_\infty \leq \epsilon}{\arg\max} \mathcal{L}_{\text{chunk-wise}}(V + \delta, Y), \quad V_{\text{adv}} = V + \delta^* \tag{5}$$

We have predefined chunk-wise labels for both anomalies and pseudo-anomalies, denoted as $Y$, which are utilized during training. The adversarial training follows a min–max optimization strategy, aiming to adjust the model parameters $\Theta$ to minimize the expected loss over adversarially perturbed data samples from the training batch $\mathcal{B}$:

$$\min_\Theta \mathbb{E}_{(V,Y)\in\mathcal{B}} \left[ \max_{\|\delta\|_\infty \leq \epsilon} \mathcal{L}_{\text{chunk-wise}}(V + \delta, Y) \right]. \tag{6}$$

**Analysis of the $\epsilon$ Value for Attack and Training.** In video anomaly detection models, the input typically consists of high-dimensional video sequences, often containing at least 100 frames, each with a resolution of 224×224 pixels. Due to the large size of these inputs, adversarial perturbations tend to be substantial, which can destabilize adversarial training Sharma and Chen [2018]. To mitigate this, some approaches like Shaeiri et al. [2020] have explored gradually increasing the value of $\epsilon$. The performance of this strategy is detailed in Appendix E. In our experiments, we adopt an $\epsilon$ value of $\frac{0.5}{255}$ as the default setting for training and evaluation. To validate this choice, we conducted an experiment on the Shanghai dataset, which provides frame-level annotations for the training set. Initially, we trained our framework on fully supervised data, representing the optimal scenario. In this setup, the model achieved near-perfect performance across both overall and anomaly-specific metrics. However, when we trained the model using higher $\epsilon$ values of $\frac{2.0}{255}$ and $\frac{1.0}{255}$, the model's standard detection performance, even without adversarial attacks, dropped to near-random levels. In contrast, with $\epsilon = \frac{0.5}{255}$, the model maintained stable training and exhibited robust performance against adversarial attacks. Further details of this experiment can be found in Table 10.

## 5 Experiments

To demonstrate the effectiveness of FrameShield, we conducted extensive experiments across several well-established benchmarks in the VAD domain. We compared our approach with various SOTA

Table 5: Frame-level detection performance (%AUROC) comparing the baseline with our proposed contributions: Pseudo Anomaly, Pseudo Label, and their combination. $AUC_O$ and $AUC_A$ represent the AUC computed on the overall test set and only on abnormal test videos, respectively, under clean and adversarial (PGD-1000, $\epsilon = \frac{0.5}{255}$) conditions.

| Method | Attack | TAD | | UCF Crime | | MSAD | |
|---|---|---|---|---|---|---|---|
| | | $AUC_O$ | $AUC_A$ | $AUC_O$ | $AUC_A$ | $AUC_O$ | $AUC_A$ |
| Pseudo Anomaly | Clean / PGD | 75.2 / 70.3 | 53.7 / 18.7 | 72.6 / 68.2 | 59.9 / 39.4 | 67.5 / 62.0 | 62.0 / 49.7 |
| Pseudo Label | Clean / PGD | 88.2 / 73.2 | 52.6 / 7.1 | 83.1 / 71.3 | 60.7 / 15.4 | 80.6 / 64.2 | 60.9 / 21.7 |
| **Pseudo Anomaly + Pseudo Label** | Clean / PGD | 85.1 / **77.2** | 50.9 / **30** | 80.2 / **78.7** | 60.1 / **53.4** | 78.9 / **76.2** | 64.4 / **60.2** |

methods, reporting AUROC metrics for both standard setups and adversarial attack scenarios. The results for the complete test sets of these benchmarks, denoted as $AUC_O$, are presented in Table 1, while the performance metrics specific to the anomaly sections of the test sets, denoted as $AUC_A$, representing a more challenging evaluation, are detailed in Table 2. These tables highlight the shortcomings of existing SOTA methods and emphasize the enhanced robustness and effectiveness of our proposed approach. A detailed comparison with recent multimodal LLM-based methods is provided in Appendix K. In Appendix L, we further evaluate *FrameShield* under black-box attack settings, and Appendix N presents comparisons with adversarially trained versions of baseline VAD methods to ensure fairness in evaluation. Collectively, all experiments consistently confirm the robustness and overall superiority of *FrameShield*.

**Analyzing Results.** As shown in Tables 1 and 2, prior SOTA methods such as UMIL, RTFM, and MGFN experience significant performance degradation under adversarial conditions, despite achieving strong results on clean data. These shortcomings motivated the development of FrameShield, our proposed solution. On average, FrameShield improves robust detection across various datasets by up to 71.0%. As highlighted in previous research, a slight reduction in clean performance is generally considered an acceptable trade-off for substantial improvements in robustness (see Appendix O for further discussion).

**Implementation Details and Dataset.** For training, we used a learning rate of $8 \times 10^{-6}$ with a chunk size of 16 frames. The model was trained for 40 epochs using the **AdamW** optimizer, which effectively incorporates weight decay. To schedule the learning rate, we applied a **Cosine scheduler**, which progressively reduces the learning rate following a cosine decay pattern. This approach promotes smoother convergence and improved generalization. We evaluate our method on well-established benchmarks: MSAD Zhu et al. [2024], UCF-Crime Sultani et al. [2018], ShanghaiTech Liu et al. [2018], TAD Lv et al. [2021], and UCSD Ped2 Mahadevan et al. [2010]; additional details are provided in Appendix A. In the adversarial scenario, we assess each method using the $l_\infty$ PGD-1000 attack with a perturbation magnitude of $\epsilon = \frac{0.5}{255}$. The evaluations under the $l_2$ norm are provided in Appendix G.

# 6   Ablation Study

In this section, we present a detailed analysis our method's component and evaluate their effectiveness.

**Ablation on Pseudo Supervision Components.** To evaluate the effectiveness of our proposed pseudo-label generator (PromptMIL) and pseudo-anomaly generator (SRD), we conduct an ablation study, as shown in Table 5. In this experiment, we train and evaluate *FrameShield* under three configurations: using only pseudo-label supervision, using only pseudo-anomaly generation, and using our default setup that incorporates both. The results demonstrate that integrating real anomaly information with precisely generated pseudo-anomaly labels substantially enhances the model's adversarial robustness.

**Video-Level Attack vs. Frame-Level Attack** In the VAD domain, most models operate at the video level, utilizing various aggregators in conjunction with MIL-based loss functions for video-level supervision. As presented in Table 4, we train our PromptMIL model under standard conditions, without adversarial training, employing different aggregators such as Max, LSE, SmoothMax, and ABMIL. Following training, we apply adversarial attacks targeting the final video-level scores of this clean model. Our experiments indicate that even the most effective gradient-flow-based aggregators are unable to degrade the model's performance to the point of zero AUC. In contrast, our frame-level

adversarial attack succeeds in entirely deceiving the model, showcasing the superior effectiveness and robustness of our proposed approach. This highlights the strategic advantage of shifting to frame-level adversarial training, enabling stronger and more impactful adversarial perturbations. .

**Advanced Attacks** We employed the PGD attack Madry et al. [2017] for both the training and evaluation phases of our model. To further demonstrate the flexibility and resilience of our proposed method under various adversarial scenarios, we also assessed its effectiveness against several advance attack strategies, including AutoAttack Croce and Hein [2020] and $A^3$ (Adversarial Attack Automation) Liu et al. [2022] in Table 4. Comprehensive details of our adaptation methods for these attack types within the VAD context are provided in Appendix I. Notably, the training process remained straightforward, consistently using the standard PGD-10 configuration to maintain simplicity and practicality.

**Additional Ablation Studies** We conducted further experiments to evaluate the impact of the SDR component, specifically analyzing the effects of Grad-CAM and Motion, as detailed in Appendix H. Additionally, we performed an ablation study on training our FrameShield model with MIL-based adversarial example generation, which is discussed in Appendix J. Furthermore, we investigated the use of alternative WSVAD methods as pseudo-label generators, with the results also presented in Appendix C.

# 7 Conclusion

We introduced FrameShield, a novel approach to enhance adversarial robustness in Weakly Supervised Video Anomaly Detection (WSVAD). Our method employs frame-level adversarial training with chunk-wise pseudo-labels generated from weakly labeled data and introduces Spatiotemporal Region Distortion (SRD) for precise frame-level anomaly labeling. We demonstrated the vulnerabilities of SOTA VAD models under adversarial attacks and bridged this gap with FrameShield, establishing a stronger defense mechanism for robust video anomaly detection in real-world scenarios.

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

# A  Dataset Details

- **MSAD**: The MSAD (Multi-Scene Anomaly Detection) dataset is a benchmark for video anomaly detection across a variety of real-world scenes. It contains a total of 920 videos, comprising 240 abnormal and 680 normal samples. The dataset is split into a training set with 480 videos (120 abnormal / 360 normal) and a test set with 440 videos (120 abnormal / 320 normal). The videos span multiple surveillance scenarios, including indoor and outdoor environments, and feature diverse anomalies such as Assault, Explosion, Fighting, Fire, and more. MSAD is designed to support both frame-level and video-level anomaly detection tasks, making it suitable for evaluating generalization across heterogeneous scenes.

- **UCF Crime**: UCF-Crime is a large-scale dataset for surveillance video analysis, comprising 1900 videos that span 13 categories of anomalous events, such as explosions, arrests, and road accidents. The training set includes video-level annotations with 800 normal and 810 abnormal videos, while the testing set provides frame-level labels for 140 normal and 150 abnormal videos. Due to computational constraints, we used only 50% of the dataset in our experiments. To ensure balanced representation, we randomly selected the samples uniformly from both normal and abnormal classes. The final dataset used for training included 410 normal and 410 abnormal videos, and the test set comprised 75 normal and 75 abnormal videos.

- **ShanghaiTech**: The ShanghaiTech dataset is a medium-scale collection of street surveillance videos captured from fixed angles, featuring 13 different background scenes and a total of 437 videos—330 normal and 107 anomalous. Originally intended for anomaly detection using only normal training data, the dataset is restructured for weakly supervised learning by incorporating 63 anomalous videos into the training set. This results in 238 training videos (63 abnormal and 175 normal) and 199 testing videos (44 abnormal and 155 normal), with both sets covering all 13 background scenes. We follow the same procedure to adapt the dataset for weak supervision.

- **TAD**: The TAD dataset consists of real-world traffic scene videos, totaling 25 hours in duration, with each video averaging 1,075 frames. It includes over seven types of road-related anomalies. The dataset is divided into a training set of 400 videos and a testing set of 100 videos, which includes 60 normal and 40 abnormal instances.

- **UCSD-Ped2**: The UCSD-Ped2 dataset is a small-scale surveillance dataset comprising 28 videos. It is traditionally used for *unsupervised* video anomaly detection (VAD), as its training set contains only normal samples. However, to enable fair evaluation of *weakly supervised* methods such as VAD-CLIP, we adopt a modified evaluation protocol inspired by recent literature Zhu et al. [2024]. Specifically, the dataset is restructured by randomly selecting six anomalous and four normal videos for training, while the remaining 18 videos (12 normal and 6 anomalous) are used for testing. This sampling process is repeated ten times, and the results are averaged to obtain stable and unbiased performance estimates. This setup allows weakly supervised models—which require video-level anomaly labels—to be consistently evaluated on Ped2.

# B  Performance Evaluation of the PromptMIL Framework

In FrameShield, we first trained PromptMIL using a Multiple Instance Learning (MIL)-based approach with fixed prompts. (We chose not to apply prompt tuning, as we believe fine-tuning CLIP reduces the benefits of prompt optimization for our task.) In the second stage, the trained PromptMIL model was used to generate pseudo labels for the training set. These pseudo labels were then employed for adversarial training, effectively transitioning the framework to a fully supervised setting by applying a chunk-level loss function.

To assess the effectiveness of this approach, we initially evaluated PromptMIL under clean (non-adversarial) conditions. As shown in Table 6, the model achieves performance comparable to existing state-of-the-art methods. It is important to note, however, that this evaluation is conducted on the test set, whereas PromptMIL is used exclusively to generate pseudo labels for the training set, for which ground-truth labels are not available.

Table 6: Frame-level detection performance (%AUROC) of our PromptMIL model trained in the first stage under clean (non-adversarial) condition. $AUC_O$ and $AUC_A$ represent the AUC computed on the overall test set and only on abnormal test videos, respectively, under clean and adversarial (PGD-1000, $\epsilon = \frac{0.5}{255}$) conditions.

| Method | Attack | TAD | | Shanghai | | MSAD | |
|---|---|---|---|---|---|---|---|
| | | $AUC_O$ | $AUC_A$ | $AUC_O$ | $AUC_A$ | $AUC_O$ | $AUC_A$ |
| **PromptMIL** | Clean / PGD | 90.3 / **3.4** | 55.7 / **2.6** | 96.4 / **2.0** | 67.1 / **4.7** | 81.2 / **0.2** | 68.5 / **3.6** |

Table 7: Frame-level detection performanc (% AUROC) with Pseudo Labels generated by various methods under clean and adversarial (PGD-1000, $\epsilon = \frac{0.5}{255}$) conditions over the entire test set ($AUC_O$).

| Method | Attack | TAD | Shanghai | MSAD |
|---|---|---|---|---|
| RTFM | Clean / PGD | 84.6 / 74.8 | 88.0 / 85.2 | 76.5 / 74.7 |
| MGFN | Clean / PGD | 86.2 / 78.1 | 89.1 / 86.8 | 79.5 / 77.3 |
| UMIL | Clean / PGD | 85.6 / 76.0 | 91.3 / 88.5 | 77.5 / 75.4 |
| **Ours** | Clean / PGD | 85.1 / **77.2** | 89.5 / **87.1** | 78.9 / **76.2** |

To further evaluate the robustness of our framework, we replaced our pseudo-anomaly generation module with those from other leading techniques and assessed performance under adversarial training scenarios. For additional details, refer to Appendix C.

## C   Alternative Pseudo-Labeling Strategies to the PromptMIL Framework

To validate the effectiveness of our PromptMIL model, we replaced our pseudo-anomaly generation module with those from other leading techniques, such as MGFN, RTFM, and UMIL, while keeping the second stage of the framework unchanged. As shown in Table 7, the performance remains largely consistent across these alternatives.

It is worth noting that PromptMIL is not considered a state-of-the-art model for pseudo-anomaly generation. However, its results are comparable to—or in some cases better than—those achieved using more advanced models. This outcome supports the hypothesis that pseudo labels generated for the training set tend to yield strong results, even when the underlying model does not generalize well to unseen test data.

## D   Analysis of Sensitivity to the Pseudo-Labeling Threshold $\tau$

We conduct a comprehensive analysis of the pseudo-labeling threshold $\tau$ used in our framework. As discussed in the paper, one of the motivations behind the SRD module is to mitigate false positives and false negatives during pseudo-label generation. The threshold $\tau$ directly governs this balance:

- Lower $\tau$ values (classifying more frames as normal) tend to increase false positives.
- Higher $\tau$ values (classifying more frames as abnormal) tend to increase false negatives.

Since our task is binary classification, setting $\tau = 0.5$ serves as a natural and well-defined decision boundary. In clean evaluation settings, model predictions are typically confident, so small changes in $\tau$ (e.g., 0.4–0.6) have minimal effect on overall performance.

Within the adversarial training framework, particularly under strong attacks such as PGD, model predictions become less confident. In these cases, false positives and false negatives during pseudo-label generation become more consequential, and the choice of $\tau$ plays a more critical role in balancing these errors. Consequently, the model's performance varies more noticeably across different threshold values under attack conditions.

All results in Table 8 correspond to the adversarially trained model. The increased sensitivity to $\tau$ under adversarial attacks reflects the amplified impact of pseudo-labeling errors in such challenging scenarios. Overall, $\tau = 0.5$ achieves the best balance between clean and adversarial performance, confirming it as a robust and principled choice within our framework.

Table 8: Sensitivity analysis of the pseudo-labeling threshold $\tau$. Results are reported as Clean / PGD for both $AUC_O$ and $AUC_A$. Performance is stable around $\tau = 0.5$, while deviations cause more variation under adversarial attacks.

| Threshold ($\tau$) | TAD | | Shanghai | | MSAD | |
|---|---|---|---|---|---|---|
| | $AUC_O$ | $AUC_A$ | $AUC_O$ | $AUC_A$ | $AUC_O$ | $AUC_A$ |
| 0.3 | 91.3 / **81.2** | 58.7 / **11.0** | 95.4 / **89.3** | 65.6 / **21.7** | 81.2 / **77.1** | 65.9 / **29.3** |
| 0.4 | 86.2 / **78.3** | 53.6 / **23.7** | 92.0 / **89.0** | 63.5 / **49.9** | 80.5 / **77.3** | 63.6 / **53.2** |
| **0.5** | 85.1 / **77.2** | 50.9 / **30.0** | 89.5 / **87.1** | 62.3 / **61.9** | 78.9 / **76.2** | 64.4 / **60.2** |
| 0.6 | 85.8 / **78.1** | 51.1 / **23.6** | 92.3 / **88.9** | 60.5 / **53.6** | 80.1 / **75.6** | 59.0 / **54.1** |
| 0.7 | 90.0 / **80.7** | 49.1 / **12.8** | 93.8 / **88.5** | 59.6 / **23.4** | 80.6 / **76.0** | 58.9 / **27.2** |

Table 9: Comparison of frame-level detection performance (%AUROC) between fixed and progressively increasing $\epsilon$ during adversarial training. Results show that gradually increasing $\epsilon$ from $\frac{0.1}{255}$ to $\frac{2.0}{255}$ does not yield significant improvement over using a fixed $\epsilon$.

| Method | Attack | TAD | | Shanghai | | MSAD | |
|---|---|---|---|---|---|---|---|
| | | $AUC_O$ | $AUC_A$ | $AUC_O$ | $AUC_A$ | $AUC_O$ | $AUC_A$ |
| **Gradually Increase** $\epsilon$ | Clean / PGD | 86.2 / **73.0** | 51.3 / **31.7** | 88.1 / **86.5** | 61.2 / **59.3** | 79.4 / **75.9** | 65.1 / **59.3** |
| **Ours** | Clean / PGD | 85.1 / **77.2** | 50.9 / **30.0** | 89.5 / **87.1** | 62.3 / **61.9** | 78.9 / **76.2** | 64.4 / **60.2** |

# E Detailed Analysis of $\epsilon$ Values During Training and Testing

In this section, we analyze the effect of different $\epsilon$ values. First, we explore a method aimed at enhancing adversarial robustness for larger $\epsilon$ values and high-dimensional data.

Shaeiri et al. [2020], building on the intuition behind weight initialization strategies in deep learning—commonly effective in various optimization scenarios Li et al. [2018]—propose a progressive approach to adversarial training. Specifically, they suggest starting with a small perturbation magnitude $\epsilon$ and gradually increasing it throughout training. As shown in Table 9, we implemented this by linearly increasing $\epsilon$ from $\frac{0.1}{255}$ to $\frac{2.0}{255}$ across epochs. However, the results indicate no significant improvement over training with a fixed $\epsilon$ value.

Next, we examine the impact of training with higher $\epsilon$ values in the context of Video Anomaly Detection (VAD). We selected $\epsilon = \frac{0.5}{255}$ for training and evaluation, as we observed that excessive perturbation magnitudes can destabilize training and severely degrade even the clean performance of the model. To validate this, we conducted an experiment on the Shanghai dataset, which includes frame-level ground truth annotations for both the training and test sets. We trained FrameShield using the real labels in a supervised setting. Under normal conditions (i.e., without attack), the model is expected to perform well. However, as shown in Table 10, training with $\epsilon = \frac{1}{255}$ or $\epsilon = \frac{2}{255}$ leads to unstable behavior and poor clean performance. In contrast, training with lower $\epsilon$ values results in stable learning and satisfactory performance both under clean and adversarial conditions, thus supporting our hypothesis.

# F Effect of Label Noise in Adversarial Training

Adversarial training based solely on pseudo-labeled data is impractical due to the inherent inaccuracies introduced by pseudo-labeling methods. These inaccuracies, typically in the form of false positives (FP) and false negatives (FN), contribute to what is commonly referred to as label noise. To mitigate this issue, we incorporate pseudo-anomalies into our training process.

In this section, we explain the effect of label noise and why we believe that label noise in the adversarial training setup can be even more detrimental.

Label noise—especially in tasks involving anomaly detection—can significantly degrade the learning process. False positives introduce normal instances that are incorrectly labeled as anomalous, while false negatives cause true anomalies to be mistakenly treated as normal. In standard supervised learning, such noise reduces classification accuracy and generalization. However, in adversarial training, the impact is amplified for the following reasons:

Table 10: Frame-level detection performance (% AUROC) of our model trained under different PGD attack strengths ($\epsilon$ values) on the fully supervised Shanghai dataset. All test-time PGD attacks in the table (bold and black numbers) are performed with $\epsilon = \frac{0.5}{255}$. $AUC_O$ and $AUC_A$ represent the AUC on the entire test set and on abnormal test videos only, respectively.

| Dataset | Training-time $\epsilon$ values | | | | | | | | | |
| --- | --- | --- | --- | --- | --- | --- | --- | --- | --- | --- |
| | $\epsilon = \frac{0}{255}$ (Clean) | | $\epsilon = \frac{0.3}{255}$ | | $\epsilon = \frac{0.5}{255}$ | | $\epsilon = \frac{1}{255}$ | | $\epsilon = \frac{2}{255}$ | |
| | $AUC_O$ | $AUC_A$ | $AUC_O$ | $AUC_A$ | $AUC_O$ | $AUC_A$ | $AUC_O$ | $AUC_A$ | $AUC_O$ | $AUC_A$ |
| Shanghai | 98.1 / **2.1** | 83.6 / **1.4** | 96.2 / **82.8** | 75.3 / **60.6** | 95.4 / **90.1** | 71.6 / **67.2** | 68.1 / **26.1** | 69.7 / **15.4** | 63.4 / **23.9** | 65.3 / **12.9** |

Table 11: Evaluation of FrameShield's robustness when trained using PGD-10 with an $\ell_\infty$ norm at $\epsilon = \frac{0.5}{255}$, and tested against PGD-1000 attacks using $\ell_2$ norms with varying $\epsilon$ values. The results indicate that the model maintains consistent robustness across different attack types.

| $\epsilon$ | TAD | | Shanghai | | MSAD | |
| --- | --- | --- | --- | --- | --- | --- |
| | $AUC_O$ | $AUC_A$ | $AUC_O$ | $AUC_A$ | $AUC_O$ | $AUC_A$ |
| Clean | 85.1 | **50.9** | 89.5 | **62.3** | 78.9 | **64.4** |
| $\frac{16}{255}$ | 79.6 | **34.2** | 85.7 | **60.0** | 76.4 | **59.9** |
| $\frac{32}{255}$ | 78.4 | **31.2** | 84.5 | **59.8** | 75.3 | **58.0** |
| $\frac{64}{255}$ | 65.8 | **20.5** | 70.4 | **51.7** | 62.9 | **49.3** |

First, consider a case where a real anomaly is mistakenly assigned a "normal" pseudo-label. Adversarial training will push this instance deeper into the "anomaly" region of the feature space. However, the loss function (e.g., cross-entropy) will penalize the model for predicting such an instance as normal—even though the label is incorrect—thus introducing contradictory signals. Conversely, if a normal sample is wrongly labeled as an anomaly, adversarial training will exaggerate its normal characteristics. The model is then simultaneously forced to treat this increasingly normal instance as "anomalous." These contradictions lead the model to overfit on noisy labels and attempt to learn a complex, unstable decision boundary.

An interesting phenomenon we observed occurs when applying a MIL-based loss function with a max-aggregator. If a false negative (i.e., an anomalous instance labeled as normal) is present in a bag (e.g., a video with many frames), adversarial training tends to amplify the anomaly traits of that instance. Consequently, the model assigns it a high anomaly score. Due to the max aggregation strategy, this high-scoring instance dominates the bag-level prediction. The model then applies the cross-entropy loss to force the prediction back toward "normal," despite the fact that the instance is truly anomalous.

As a result, during training, false negatives are consistently selected during the aggregation step due to the effect of adversarial perturbations. The loss is therefore repeatedly computed on incorrectly labeled samples. This persistent misalignment between the label and the actual instance leads the MIL loss under adversarial training to fail to train properly, ultimately preventing the model from converging when noisy labels are present.

## G Robustness Evaluation Against PGD Attacks with $\ell_2$ Norm

To evaluate the robustness of our model, we conducted an experiment in which *FrameShield* was trained and tested using PGD attacks under the same $\ell_\infty$ norm with varying $\epsilon$ values. Specifically, the $\epsilon$ used during training matched the one used during evaluation. As shown in Table 11, the results demonstrate that *FrameShield* maintains strong performance across a range of $\epsilon$ values, indicating its robustness to different levels of adversarial perturbation.

## H Component-wise Ablation Study of the SRD Module

In this section, we assess the effectiveness of the individual components of our SRD module. As shown in Table 12, we first replace the Grad-CAM-based foreground selection with a baseline that

Table 12: Comparison between our Grad-CAM-based foreground selection and a random region baseline. The results demonstrate that using Grad-CAM significantly improves foreground localization, validating its effectiveness within the SRD module.

| Method | Attack | TAD | | Shanghai | | MSAD | |
|---|---|---|---|---|---|---|---|
| | | $AUC_O$ | $AUC_A$ | $AUC_O$ | $AUC_A$ | $AUC_O$ | $AUC_A$ |
| **Random Region Distortion** | Clean / PGD | 81.0 / **66.2** | 51.6 / **19.8** | 84.2 / **78.3** | 60.4 / **47.5** | 73.6 / **65.9** | 60.7 / **47.1** |
| **Ours (Grad-CAM)** | Clean / PGD | 85.1 / **77.2** | 50.9 / **30.0** | 89.5 / **87.1** | 62.3 / **61.9** | 78.9 / **76.2** | 64.4 / **60.2** |

Table 13: Performance comparison of our curved vector-based motion approach with two alternatives: no motion and random location transfer. Our method consistently outperforms both, highlighting the importance of structured temporal transformations in anomaly modeling.

| Method | Attack | TAD | | Shanghai | | MSAD | |
|---|---|---|---|---|---|---|---|
| | | $AUC_O$ | $AUC_A$ | $AUC_O$ | $AUC_A$ | $AUC_O$ | $AUC_A$ |
| **No Motion** | Clean / PGD | 77.9 / **53.2** | 50.3 / **9.7** | 82.5 / **72.4** | 60.2 / **28.8** | 71.6 / **58.9** | 59.9 / **31.2** |
| **Random Location Transfer** | Clean / PGD | 82.3 / **64.1** | 50.3 / **21.8** | 82.1 / **75.2** | 61.4 / **45.8** | 70.2 / **62.7** | 61.8 / **43.5** |
| **Ours** | Clean / PGD | 85.1 / **77.2** | 50.9 / **30.0** | 89.5 / **87.1** | 62.3 / **61.9** | 78.9 / **76.2** | 64.4 / **60.2** |

selects a random region in the frame and applies a curved vector in a random direction. The results demonstrate the clear advantage of our method in identifying foreground objects. Nonetheless, we acknowledge that Grad-CAM is not an optimal solution; in future work, the foreground detection component could be enhanced using more advanced models, such as pretrained object detectors.

To further assess the contribution of temporal modeling, we conducted additional ablation studies. In these experiments, we replaced the curved vector-based motion with two alternative strategies: (1) No Motion, and (2) Random Location Transfer. In the first setting, all frames were distorted in the same region without any motion, but with newly applied harsh augmentations. In the second setting, after distorting a region in the first frame of a sequence, we randomly selected a position in the subsequent frame and pasted the distorted region from the first frame onto it, again applying a new set of harsh augmentations. As shown in Table 13, our method outperforms both alternatives, highlighting the effectiveness of our temporal modeling strategy.

# I  Detail of adaptation methods for various attack

Adversarial attacks were initially introduced for classification tasks. These attacks involve adding small, imperceptible perturbations to the input data of a neural network to increase its loss function. In classification, the input is typically a single image, and the output consists of logits corresponding to each class, representing the probability of the input being classified into a specific category. One of the most widely adopted and well-established attacks in this domain is **PGD-1000**, which we thoroughly describe in Section 3, including its adapted version for VAD. Subsequently, researchers have proposed even more powerful attacks to further challenge model robustness—among the most notable are **AutoAttack** and $\mathbf{A^3}$ (Adaptive AutoAttack). We also adapt these attacks to suit the VAD setting by reinterpreting their classification-based evaluation strategies. In such settings, models output class logits, and attacks typically optimize losses like cross-entropy or DLR. However, these formulations do not directly translate to Video Anomaly Detection (VAD), where models output continuous, chunk-wise anomaly scores rather than class probabilities. To adapt these attacks to VAD, we define a task-specific loss function:

$$\mathcal{L}_{\text{VAD}}(V) = \sum_{m=1}^{M} Y_t \cdot S_t(F_\Theta; V) \tag{7}$$

Here $Y_t \in \{-1, +1\}$ is the attack direction label: $+1$ for normal chunks (to increase score), $-1$ for abnormal chunks (to decrease score), and $m$ is the total number of chunks. We adapted AutoAttack by replacing the cross-entropy and DLR losses in its APGD and FAB components with $\mathcal{L}_{\text{VAD}}(V)$. The DLR loss, which assumes at least three output logits, was removed entirely due to its incompatibility with scalar outputs. The Square Attack component remained unchanged, as it operates independently of the loss formulation and is inherently compatible with score-based tasks like VAD. The $A^3$ framework was also adapted for the VAD setting. Unlike AutoAttack, which combines multiple attack methods, $A^3$ is a standalone attack strategy that enhances robustness evaluation through two core mechanisms: **Adaptive Direction Initialization** (ADI) and **Online Statistics-based Discarding**

Table 14: Adversarial training results of PromptMIL using video-level perturbations generated by a MIL-based attack with a Max aggregator. The results indicate no significant improvement in robustness, underscoring the weakness of this adversarial training strategy.

| Method | Attack | TAD | | Shanghai | | MSAD | |
|---|---|---|---|---|---|---|---|
| | | $AUC_O$ | $AUC_A$ | $AUC_O$ | $AUC_A$ | $AUC_O$ | $AUC_A$ |
| **MIL-Base Loss Adv Training** | Clean / PGD | 86.2 / **26.1** | 52.7 / **5.9** | 87.3 / **37.1** | 59.7 / **15.2** | 79.1 / **26.6** | 65.0 / **11.3** |
| **Ours** | Clean / PGD | 85.1 / **77.2** | 50.9 / **30.0** | 89.5 / **87.1** | 62.3 / **61.9** | 78.9 / **76.2** | 64.4 / **60.2** |

(OSD). To apply $A^3$ in the context of video anomaly detection, we retained our custom VAD-specific loss function—used also in the PGD adaptation—that manipulates anomaly scores to degrade detection performance. The ADI component accelerates convergence by learning model-specific perturbation directions from successful restarts and reusing them to generate more effective adversarial examples. Meanwhile, OSD improves efficiency by identifying and discarding hard-to-attack video segments during the attack process, thereby reallocating computational effort toward easier-to-attack samples. These two mechanisms work together within $A^3$ to generate adversarial videos that effectively degrade a model's ability to distinguish between normal and abnormal chunks under a limited computational budget.

## J Training FrameShield with MIL-based Loss

As shown in Table 3, video-level adversarial attacks based on the MIL framework are relatively weak and ineffective. To further investigate this limitation, we train our PromptMIL model using a MIL-based loss function and adversarial examples generated through video-level perturbations, utilizing a Max aggregator. As reported in Table 14, this adversarial training strategy fails to enhance model robustness, further confirming the inadequacy of using such weak adversarial examples for training.

## K Comparison with Multimodal LLM-based Methods

Recent advances in video anomaly detection (VAD) have introduced multimodal large language model (LLM)-based approaches such as Holmes-VAU Zhang et al. [2024] and LAVAD Zanella et al. [2024], which integrate visual encoders with pretrained language models for video semantic understanding. These methods can exhibit stronger general robustness to low-level pixel perturbations than traditional MIL or feature aggregation models. To investigate whether such video-understanding-based approaches are also vulnerable to adversarial perturbations, we conducted targeted adversarial attacks on both Holmes-VAU and LAVAD under conditions consistent with our evaluation setup.

Both Holmes-VAU and LAVAD employ captioning-based pipelines in which visual encoders produce embeddings that are then decoded into textual descriptions by LLMs. Specifically, Holmes-VAU uses InternVL2-2B, combining the InternViT-300M image encoder with InternLM2-Chat-1.8B as the language model. LAVAD adopts BLIP-2, which integrates a pretrained CLIP image encoder, a Q-former, and OPT-6.7B as the text decoder.

**Adversarial Attack Setup.** We treated each captioning model as an end-to-end system and applied PGD-1000 attacks to generate adversarial examples designed to alter the generated captions. For Holmes-VAU, we uniformly sampled 12 frames from each video and used a prompt such as: *"Could you specify the anomaly events present in the video?"* Under adversarial perturbations, we enforced target captions to induce misclassification. For normal videos, we forced the model to output: *"There is an anomaly in the video; two cars have an accident."* For anomalous videos, we guided the model to output: *"There is no abnormal event; everything goes normal."*

To adapt the attack to the autoregressive decoding process of LLMs, the target captions were injected into the context window rather than allowing the model to condition on its own previous tokens. A similar procedure was applied to LAVAD, where adversarial frames were fed to the BLIP-2 model to produce misleading captions subsequently used for downstream anomaly scoring.

**Results and Analysis.** We evaluated both models under PGD-1000 attacks on the UCF-Crime and ShanghaiTech datasets and compared them with FrameShield. Results are summarized in Table 15.

Table 15: Comparison with multimodal LLM-based video-understanding methods under PGD-1000 attack. Results are reported as Clean / PGD $AUC_O$ (%). FrameShield maintains high robustness, while Holmes-VAU and LAVAD exhibit substantial degradation under attack.

| Method | Shanghai | UCF-Crime |
|---|---|---|
| Holmes-VAU | 95.6 / **16.0** | 88.9 / **14.2** |
| LAVAD | 91.0 / **21.4** | 80.3 / **18.7** |
| **FrameShield (Ours)** | 89.5 / **87.1** | 80.2 / **78.7** |

Table 16: Evaluation of FrameShield under white-box (PGD-1000) and black-box (NES, Bandit) attack settings. Results are reported as $AUC_O$ / $AUC_A$ (%). FrameShield maintains strong robustness across both threat models.

| Attack | TAD | UCF-Crime | MSAD |
|---|---|---|---|
| Clean | 85.1 / **50.9** | 80.2 / **60.1** | 78.9 / **64.4** |
| PGD-1000 (White-box) | 77.2 / **30.0** | 78.7 / **53.4** | 76.2 / **60.2** |
| NES (Black-box) | 84.2 / **44.5** | 79.8 / **58.1** | 78.4 / **64.1** |
| Bandit (Black-box) | 83.5 / **43.9** | 79.3 / **57.7** | 77.9 / **63.8** |

Although multimodal LLM-based methods show stronger baseline robustness than traditional MIL-based approaches, both are still highly susceptible to strong, targeted adversarial perturbations. Notably, Holmes-VAU—processing 12 frames jointly—was more vulnerable than LAVAD due to its higher input dimensionality. FrameShield achieved significantly higher robustness while maintaining competitive clean performance.

These results indicate that although multimodal LLM-based frameworks like Holmes-VAU and LAVAD possess enhanced semantic understanding and resilience to benign noise, they remain highly vulnerable to carefully crafted adversarial perturbations. FrameShield consistently outperforms both methods under attack, highlighting its effectiveness in preserving robustness against adversarial manipulations targeting both visual and semantic cues.

## L  Evaluating FrameShield in the Black-box Setup

To further assess the robustness of *FrameShield* beyond the white-box setting, we evaluate its performance under *black-box* adversarial attacks. Unlike white-box attacks—where the adversary has full access to model parameters and gradients—black-box attacks assume limited knowledge, relying only on model queries to estimate gradients. Although white-box attacks are less common in real-world applications, they provide a rigorous benchmark for stress-testing robustness. Importantly, models that demonstrate resilience in the white-box setting often maintain robustness against weaker black-box perturbations.

To validate this hypothesis, we conducted black-box experiments using two representative query-based attack algorithms: the Natural Evolution Strategy (NES) Ilyas et al. [2018] and the Bandit attack Ilyas et al. [2019]. Both methods iteratively approximate gradients through queries without direct access to model internals.

Table 16 summarizes the results on three representative benchmarks. As expected, FrameShield exhibits smaller performance degradation under black-box settings compared to white-box PGD-1000 attacks, confirming that robustness acquired through adversarial training effectively transfers to more realistic threat models. These findings demonstrate that FrameShield maintains strong resistance to both white-box and black-box adversarial perturbations.

## M  More Detailed Related Work

Video anomaly detection (VAD) is a critical computer vision task with real-world applications in manufacturing, healthcare, and public safety, where detecting abnormal events such as accidents, fights, or equipment failure can mitigate risks and prevent losses. However, annotating anomaly data at the frame level is extremely expensive and time-consuming due to the rare and ambiguous nature

of anomalous events. This challenge has motivated two dominant research paradigms: unsupervised methods, which learn only from normal data, and weakly-supervised methods, where only video-level anomaly labels are provided without precise temporal annotations. In recent years, the weakly supervised video anomaly detection (WSVAD) setting has gained significant traction due to its good balance between annotation burden and detection performance.

Most SOTA WSVAD methods leverage the Multiple Instance Learning (MIL) framework, treating each video as a bag of instances (snippets) under the assumption that at least one snippet in an anomalous video exhibits abnormal behavior. Although recent WSVAD methods have achieved near-perfect performance on standard metrics, their vulnerability to adversarial attacks remains a critical issue that threatens their reliability in real-world deployments.

Current WSVAD models predominantly rely on frozen pretrained architectures such as I3D, C3D, SwinTransformer, and CLIP, which are originally trained on large-scale datasets. These models serve to reduce dimensionality and extract meaningful embeddings for anomaly detection. However, they are inherently susceptible to adversarial manipulations, as demonstrated by Schlarmann et al. [2024], Chen et al. [2019], Li et al. [2021], indicating a substantial gap in adversarial robustness within the WSVAD landscape.

The introduction of WSVAD as a MIL-based problem began with Sultani et al. [2018], who proposed a large-scale dataset alongside a straightforward yet effective MIL-based method that relies on selecting snippets that look most suspicious. Despite its effectiveness, standard MIL approaches often suffer from label noise and limited temporal granularity, which can impair detection accuracy.

To address these issues, Zhong et al. [2019] reformulated WSVAD as a binary classification problem with noisy labels and employed a graph convolutional network (GCN) to suppress noise. While effective, this approach is computationally intensive and can produce an unconstrained feature space. To overcome these limitations, Tian et al. [2021] proposed RTFM, which learns from feature magnitudes—encouraging higher magnitudes for abnormal snippets—and introduced a Multi-scale Temporal Network (MTN) to model both short- and long-range temporal dependencies, enhancing robustness against noisy frames. Subsequently, MGFN Chen et al. [2022] improved the modeling of temporal relations by employing a transformer-based glance-and-focus mechanism with a contrastive loss to better distinguish between normal and abnormal patterns. Additionally, UR-DMU Zhou et al. [2023] introduced dual memory units and uncertainty modeling to better distinguish between normal and anomalous data. Further advancements include UMIL Lv et al. [2023] , which addresses the bias in traditional MIL by proposing an unbiased training framework that leverages both confident and ambiguous snippets. It applies feature-space clustering to identify latent pseudo-labels for uncertain snippets and incorporates them into the training using contrastive loss and end-to-end fine-tuning.

A recent wave of research explores the integration of Vision-Language Models (VLMs) like CLIP for WSVAD. These models leverage semantic richness and textual understanding to enhance anomaly detection. For instance, UMIL utilizes CLIP as a feature extracto. CLIP-TSA Joo et al. [2023] employs CLIP as a visual feature extractor and models long- and short-range temporal dependencies through Temporal Self-Attention (TSA). Additionally, CLIP-VAD Wu et al. [2024] enhances anomaly detection by incorporating extra supervision for different types of anomalies in videos (e.g., abuse, arrest, assault, etc.) along with learnable prompts. It also integrates a Local-Global Temporal Adapter (LGT-Adapter) to effectively capture both short-term and long-term dependencies. Furthermore, TEVAD Chen et al. [2023] proposes the use of SwinBERT (vulnerable backbone) for video captioning to enhance semantic understanding. It then fuses these caption-based features with I3D-extracted visual features through Multi-Scale Temporal Networks (MTN).

Overall, the evolution of WSVAD methods reflects a shift towards more sophisticated temporal modeling, enhanced feature extraction, and the integration of multi-modal approaches like vision-language models. However, despite these advancements, the **reliance on vulnerable feature extractors** and the **lack of real frame-level labels** hinder the practical application of adversarial training, leaving these models notably unrobust and vulnerable to attacks.

## N   Adversarial Training of Baseline Methods

To ensure a fair comparison, we additionally adversarially trained several baseline methods using the same adversarial training protocol as FrameShield. While the original versions of these models were

Table 17: Comparison of adversarially trained baselines under clean and PGD-1000 attack settings. Results are reported as Clean / PGD AUC (%). FrameShield maintains superior robustness while preserving competitive clean performance.

| Method | Attack | Shanghai | | TAD | | MSAD | |
|---|---|---|---|---|---|---|---|
| | | $AUC_O$ | $AUC_A$ | $AUC_O$ | $AUC_A$ | $AUC_O$ | $AUC_A$ |
| RTFM | Clean / PGD | 88.7 / **17.3** | 63.1 / **6.5** | 83.4 / **16.0** | 51.0 / **8.0** | 80.1 / **21.6** | 67.9 / **3.2** |
| VAD-CLIP | Clean / PGD | 93.1 / **15.4** | 61.5 / **9.7** | 88.4 / **14.2** | 53.9 / **10.3** | – / **–** | – / **–** |
| Base MIL | Clean / PGD | 89.0 / **17.2** | 61.2 / **4.9** | 82.9 / **14.3** | 48.5 / **3.3** | 77.6 / **19.8** | 60.5 / **2.1** |
| UMIL | Clean / PGD | 91.4 / **18.3** | 63.3 / **9.0** | 87.8 / **19.7** | 49.7 / **7.5** | 78.8 / **23.1** | 66.7 / **4.5** |
| **FrameShield (Ours)** | Clean / PGD | 89.5 / **87.1** | 62.3 / **61.9** | 85.1 / **77.2** | 50.9 / **30.0** | 78.9 / **76.2** | 64.4 / **60.2** |

trained only under standard conditions, this extension allows a direct evaluation of how well existing approaches adapt to adversarial optimization. The results—covering both clean and adversarial (PGD) performance—are summarized in Table 17.

Across all datasets, the adversarially trained baselines exhibit substantial degradation in clean accuracy and limited improvement in robustness. In contrast, FrameShield maintains strong performance under both clean and adversarial conditions. This discrepancy can be attributed to several key limitations in prior works. Methods such as RTFM and VAD-CLIP employ frozen feature extractors (e.g., I3D or CLIP), preventing the backbone from adapting to adversarial perturbations during training. MIL-based models like Base MIL and UMIL rely on hard MAX temporal aggregation, which restricts gradient propagation and undermines the effectiveness of adversarial optimization. Moreover, approaches such as UMIL are highly sensitive to noise in pseudo-labels, and this issue is further amplified under adversarial perturbations.

FrameShield addresses these limitations through three main design strategies. First, the model is trained in a fully end-to-end manner, allowing the backbone to adapt to adversarially perturbed data. Second, frame-level binary pseudo labels replace unstable MAX aggregation, resulting in smoother gradient flow and more stable optimization. Finally, the synthetic region disturbance (SRD) module introduces controlled perturbations that reduce the impact of false positives and false negatives, improving robustness against label noise. Together, these design choices enable FrameShield to achieve both high clean accuracy and strong adversarial robustness, significantly outperforming other methods even when they are adversarially trained.

# O    Discussion on Clean vs. Adversarial Trade-off

A common concern in adversarial robustness research is the trade-off between clean and adversarial performance. We provide additional clarification and empirical evidence here. While FrameShield shows a reduction in clean accuracy under adversarial training, this outcome is consistent with a well-established phenomenon: improving adversarial robustness often comes at the cost of reduced clean performance. This trade-off is not a limitation unique to our approach but is intrinsic to robust optimization in general. As noted in the limitations section, the tension between clean accuracy and robustness has been extensively studied in prior work Tsipras et al. [2019]. For instance, on ImageNet, the robust variant of ResNet-50 has been reported to drop from 75.8% to 65.8% in clean conditions Singh et al. [2023], emphasizing that such declines are expected and not indicative of design flaws. Importantly, when FrameShield is trained without adversarial perturbations, it demonstrates competitive clean accuracy, validating its effectiveness under standard training conditions. This outcome underscores that the observed trade-off is an inherent characteristic of adversarial training methodologies rather than a limitation of the FrameShield technique. The results are summarized in Table 18. Overall, we emphasize that this trade-off is a natural and widely recognized consequence of adversarial training, not a sign of underperformance.

**Experiments with TRADES Loss.**    We also explored alternative adversarial training objectives to better balance clean and robust performance. In particular, we incorporated the TRADES loss Zhang et al. [2019b] into FrameShield. TRADES explicitly separates natural and boundary losses, encouraging the model to align predictions on adversarially perturbed frames with those on original frames via a cross-entropy consistency term.

Table 18: AUC$_O$ scores (%) under clean and PGD adversarial settings. Results are reported as *Clean / PGD*. FrameShield (ours) demonstrates strong robustness while maintaining competitive clean performance.

| Method | Attack | UCSD Ped2 | Shanghai | TAD | UCF Crime | MSAD |
|---|---|---|---|---|---|---|
| RTFM | Clean / PGD | 98.6 / 2.4 | 97.2 / 8.5 | 89.6 / 6.3 | 85.7 / 7.3 | 86.6 / 10.0 |
| UMIL | Clean / PGD | 94.2 / 6.9 | 96.8 / 2.8 | 92.9 / 3.0 | 86.8 / 4.7 | 83.8 / 6.1 |
| VAD-CLIP | Clean / PGD | 98.4 / 6.3 | 97.5 / 3.6 | 92.7 / 5.1 | 88.0 / 8.2 | – / – |
| **Ours (Adv. trained)** | Clean / PGD | 97.1 / **81.3** | 89.5 / **87.1** | 85.1 / **77.2** | 80.2 / **78.7** | 78.9 / **76.2** |
| **Ours (Clean trained)** | Clean / PGD | 98.6 / **8.0** | 97.4 / **5.4** | 93.4 / **7.4** | 87.8 / **9.3** | 86.3 / **8.6** |

Table 19: Comparison between FrameShield trained with the standard adversarial loss and the TRADES loss. Results are reported as Clean / PGD. While TRADES improves clean accuracy slightly, it reduces robustness under PGD attack.

| Method | Attack | Shanghai | | TAD | | UCF Crime | | MSAD | |
|---|---|---|---|---|---|---|---|---|---|
| | | $AUC_O$ | $AUC_A$ | $AUC_O$ | $AUC_A$ | $AUC_O$ | $AUC_A$ | $AUC_O$ | $AUC_A$ |
| **Ours (Adapted TRADES Loss)** | Clean / PGD | 93.2 / **78.9** | 64.1 / **32.7** | 90.1 / **61.5** | 55.1 / **15.2** | 82.4 / **67.1** | 60.8 / **25.6** | 80.5 / **64.8** | 67.8 / **34.7** |
| **Ours (Default Loss)** | Clean / PGD | 89.5 / **87.1** | 62.3 / **61.9** | 85.1 / **77.2** | 50.9 / **30.0** | 80.2 / **78.7** | 60.1 / **53.4** | 78.9 / **76.2** | 64.4 / **60.2** |

Table 19 summarizes the results. While the TRADES loss led to slightly improved clean accuracy compared to standard adversarial training, it also caused a noticeable drop in robustness across all datasets. This aligns with observations in prior research that TRADES can provide smoother decision boundaries but sometimes reduces robustness under strong perturbations. Overall, FrameShield's default loss yields a more favorable balance for video anomaly detection, maintaining higher robustness while remaining competitive in clean conditions.

## P    Implementation Details

We conducted adversarial training for 40 epochs using the AdamW optimizer with a learning rate of $8 \times 10^{-6}$, a chunk size of 16 frames and $\epsilon = \frac{0.5}{255}$. A cosine scheduler was employed to gradually decrease the learning rate. We conducted our experiments on 2 NVIDIA GeForce RTX 4090 GPUs (24 GB), with the pipeline completing in approximately 30 hours. Additionally, to train PromptMIL with X-Clip Ma et al. [2022] as a feature extractor and get pseudo-labels, we required approximately 4 hours.

## Q    Detailed Results

Table 20 summarizes the performance of our method (Mean ± STD %) under both clean and adversarial conditions across five video anomaly detection datasets: UCSD-Ped2, ShanghaiTech, TAD, UCF Crime, and MSAD. The standard deviation is computed over five runs with different random seeds. The consistently low variance observed across all datasets and evaluation scenarios demonstrates the robustness and reliability of our approach.

## R    Discussion on Foreground Detection in the SRD Module

A common concern regarding SRD is whether Grad-CAM, which we employ in the SRD module, is an optimal choice for foreground detection. Alternative methods such as object detectors or attention maps (e.g., from DINOv2) might appear better suited for identifying meaningful regions. We address this concern below with additional analysis and experiments. First, we acknowledge that Grad-CAM is not a dedicated foreground detection tool. This limitation is explicitly noted in the main paper. However, its use in the SRD module is motivated by its simplicity, interpretability, and compatibility with our adversarial training framework.

**Experiments with Alternative Localizers.**    To evaluate the importance of localization, we substituted Grad-CAM with more semantically grounded approaches: DINO attention maps Caron et al. [2021], YOLO Tian et al. [2025], and Fast R-CNN Ren et al. [2016]. All other SRD components—including motion trajectory modeling, temporal coherence, and perturbation generation—were

Table 20: Frame-level detection performance (Mean ± STD %) under clean and adversarial conditions across selected datasets over the entire test videos ($AUC_O$).

| Statistics | Eval Type | UCSD-Ped2 | Shanghai | TAD | UCF | MSAD |
|---|---|---|---|---|---|---|
| Mean ± STD | Clean | $97.1 \pm 1.2$ | $89.5 \pm 1.7$ | $85.1 \pm 0.7$ | $80.2 \pm 1.3$ | $78.9 \pm 1.6$ |
| | Adv | $81.3 \pm 0.9$ | $87.1 \pm 2.0$ | $77.2 \pm 1.4$ | $78.7 \pm 0.8$ | $76.2 \pm 1.8$ |

Table 21: Comparison of different foreground detection methods under clean and PGD settings. Values are reported as Clean / PGD. Results show that while stronger localizers (YOLO, Fast R-CNN, DINO) yield comparable performance, the SRD module's temporal modeling and perturbation design play a more critical role than localization precision.

| Method | Attack | TAD | | Shanghai | | MSAD | |
|---|---|---|---|---|---|---|---|
| | | $AUC_O$ | $AUC_A$ | $AUC_O$ | $AUC_A$ | $AUC_O$ | $AUC_A$ |
| YOLO | Clean / PGD | 85.7 / 75.2 | 51.2 / 31.2 | 90.3 / 88.4 | 60.7 / 59.8 | 76.8 / 74.3 | 63.8 / 58.9 |
| Fast R-CNN | Clean / PGD | 82.6 / 73.5 | 48.3 / 26.7 | 87.7 / 84.2 | 58.9 / 59.2 | 74.2 / 71.8 | 64.1 / 60.7 |
| DINO Attention | Clean / PGD | 86.7 / 78.1 | 52.3 / 33.1 | 88.3 / 86.5 | 60.8 / 60.3 | 79.1 / 75.9 | 64.5 / 59.8 |
| **Ours (Grad-CAM)** | Clean / PGD | 85.1 / **77.2** | 50.9 / **30.0** | 89.5 / **87.1** | 62.3 / **61.9** | 78.9 / **76.2** | 64.4 / **60.2** |

kept unchanged, allowing us to isolate the effect of localization. The results are summarized in Table 21. Across all experiments, replacing Grad-CAM with YOLO, Fast R-CNN, or DINO attention did not yield significant improvements. This suggests that the SRD module's strength lies more in its perturbation generation and temporal modeling than in precise foreground detection.

# S    Pseudo-Anomaly Examples

In this section, we present examples of pseudo-anomalous data generated by our SRD module, along with their corresponding Grad-CAM visualizations. In Figure 4, we illustrate the distortion process across a sequence of frames. The top row shows the original (normal) frame sequence and the Grad-CAM heatmap of the first frame, while the bottom row displays the corresponding distorted (anomalous) frames.

# T    Pseudocode of FrameShield

We present the pseudocode in Algorithm 1 for our FrameShield framework, which comprises two sequential phases: (1) weakly supervised training using the proposed PromptMIL formulation, and (2) fully supervised adversarial training using pseudo-anomalies generated by our SRD (Spatiotemporal Region Distortion) module.

**Algorithm 1** FrameShield: Robust Video Anomaly Detection

---

**Require:** Training set of videos $\mathcal{D} = \{(V_i, y_i)\}$ with video-level labels, pretrained X-Clip model, text prompts $t_n$ ("Normal") and $t_a$ ("Abnormal")

**Ensure:** Robust anomaly detector

1: **// Phase 1: PromptMIL Training**
2: **for** each video $V_i$ in $\mathcal{D}$ **do**
3:     Partition $V_i$ into $m$ chunks: $V_i = \{v_1, v_2, ..., v_m\}$
4:     **for** each chunk $v_j$ **do**
5:         Extract feature: $f_j \leftarrow F_\Theta(v_j)$
6:         Compute dot products: $s_j^a \leftarrow f_j \cdot t_a, s_j^n \leftarrow f_j \cdot t_n$
7:         Compute softmax: $S_j \leftarrow \text{softmax}(s_j^a, s_j^n)$
8:     **end for**
9:     Aggregate anomaly score: $S_i \leftarrow \max_j S_j$
10:     Compute loss: $L_i \leftarrow \text{BCE}(S_i, y_i)$
11: **end for**
12: Update model $F_\Theta$ using total loss $\sum_i L_i$

13: **// Generate Pseudo-Labels**
14: **for** each abnormal video in $\mathcal{D}$ **do**
15:     Recompute $S_j$ for each chunk using trained $F_\Theta$
16:     Assign pseudo-labels: $\hat{y}_j \leftarrow \mathbb{1}[S_j > \tau]$ (thresholded)
17: **end for**

18: **// Phase 2: Adversarial Training with SRD**
19: **for** each normal video $V$ **do**
20:     **if** random() $< p_\text{SRD}$ **then** {With probability $p_\text{SRD}$, generate a pseudo-anomaly using SRD}
21:         Generate SRD pseudo-anomalies $\tilde{V}$:
22:             Select random frame sequence; apply Grad-CAM to locate salient region
23:             Apply mask and augmentations to create spatial distortions
24:             Introduce motion trajectory for temporal distortion
25:         Assign label $\hat{y}_\text{SRD} \leftarrow 1$
26:     **end if**
27: **end for**
28: Merge real videos with pseudo-labeled and SRD-augmented data

29: **for** each video $V$ with chunk-wise labels $Y = \{y_1, ..., y_m\}$ **do**
30:     **for** each chunk $v_j$ **do**
31:         Compute anomaly score $S_j \leftarrow F_\Theta(v_j)$
32:         Compute loss: $L_j \leftarrow \text{BCE}(S_j, y_j)$
33:     **end for**
34:     Total loss $L_V \leftarrow \sum_j L_j$
35:     Generate adversarial perturbation $\delta^* \leftarrow \arg\max_{\|\delta\|_\infty \le \epsilon} L_V(V + \delta, Y)$
36:     Update $F_\Theta$ with $(V + \delta^*, Y)$ using min-max optimization
37: **end for**

---

# U  Limitations

**Clean Performance in Video Anomaly Detection** This work focuses on enhancing the robustness of video anomaly detection models against adversarial attacks. While our approach shows notable gains in adversarial detection, its performance on clean (non-adversarial) data remains below that of current SOTA methods. This reflects the well-known trade-off between clean and adversarial performance, as highlighted in prior studies Zhang et al. [2019b]; Tsipras et al. [2019]; Madry et al. [2018] and Raghunathan et al. [2020].

**Using Grad-CAM for Object Localization** While Grad-CAM serves as a practical tool for identifying salient regions in static frames, its use within SRD brings certain inherent limitations. Grad-CAM is a gradient-based visualization technique developed primarily for classification models and does not explicitly model objectness or spatial boundaries. As a result, highlighted regions may be diffuse or

imprecise, potentially including background clutter or missing parts of coherent objects. Furthermore, since Grad-CAM relies on the internal feature activations of a pre-trained network like ResNet18 He et al. [2016], its saliency maps reflect class-discriminative attention rather than true object localization. This can lead to suboptimal or inconsistent masks, especially in complex or low-saliency scenes. Employing dedicated object detectors in place of Grad-CAM could yield more accurate and semantically meaningful regions, improving both the realism and control of the generated anomalies.

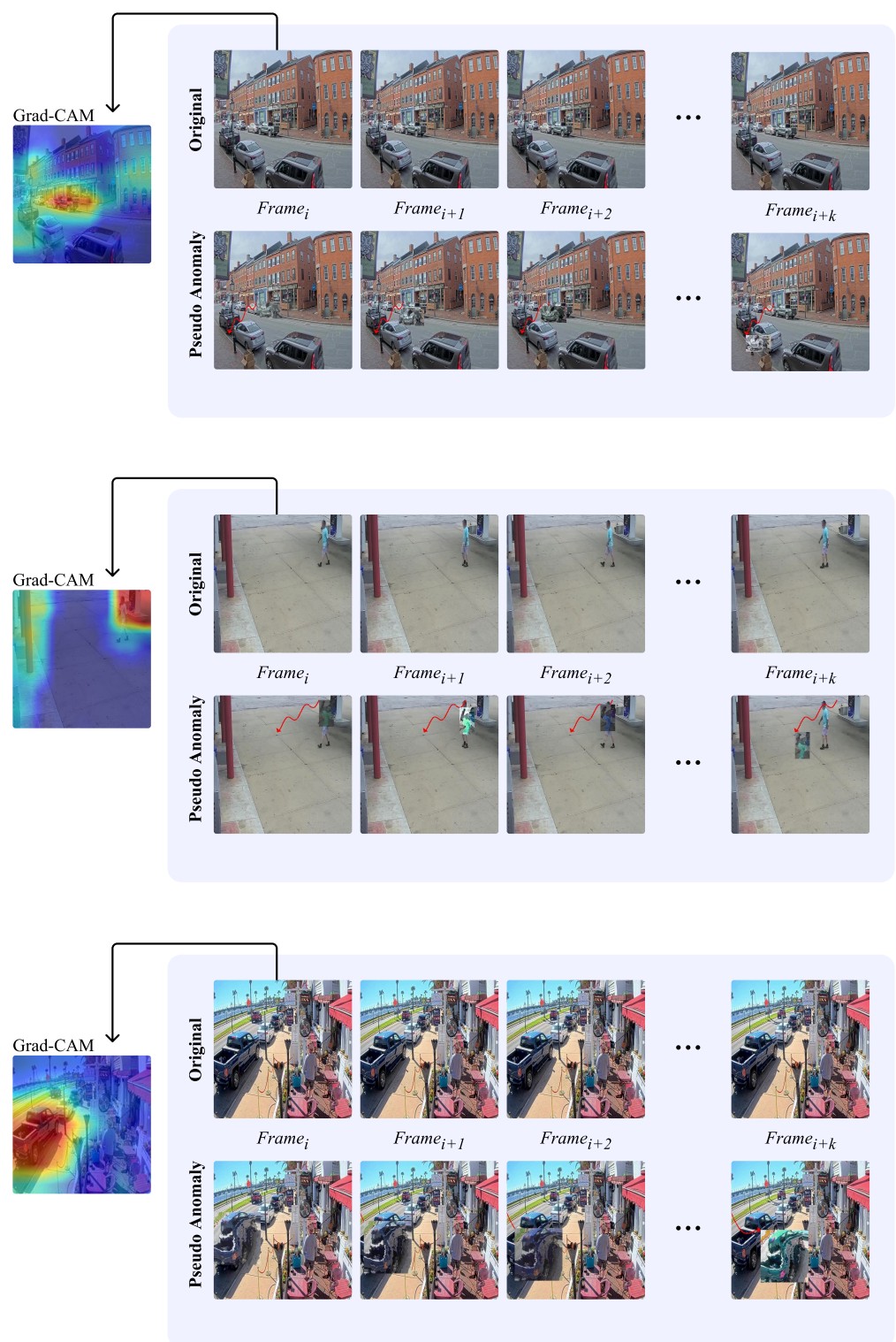

Figure 4: Visualization of pseudo-anomalous data generated by the SRD module. The top row shows the original (normal) sequence of frames along with the Grad-CAM heatmap of the first frame. The bottom row displays the corresponding distorted (pseudo-anomalous) frames created by applying the SRD distortion strategy.

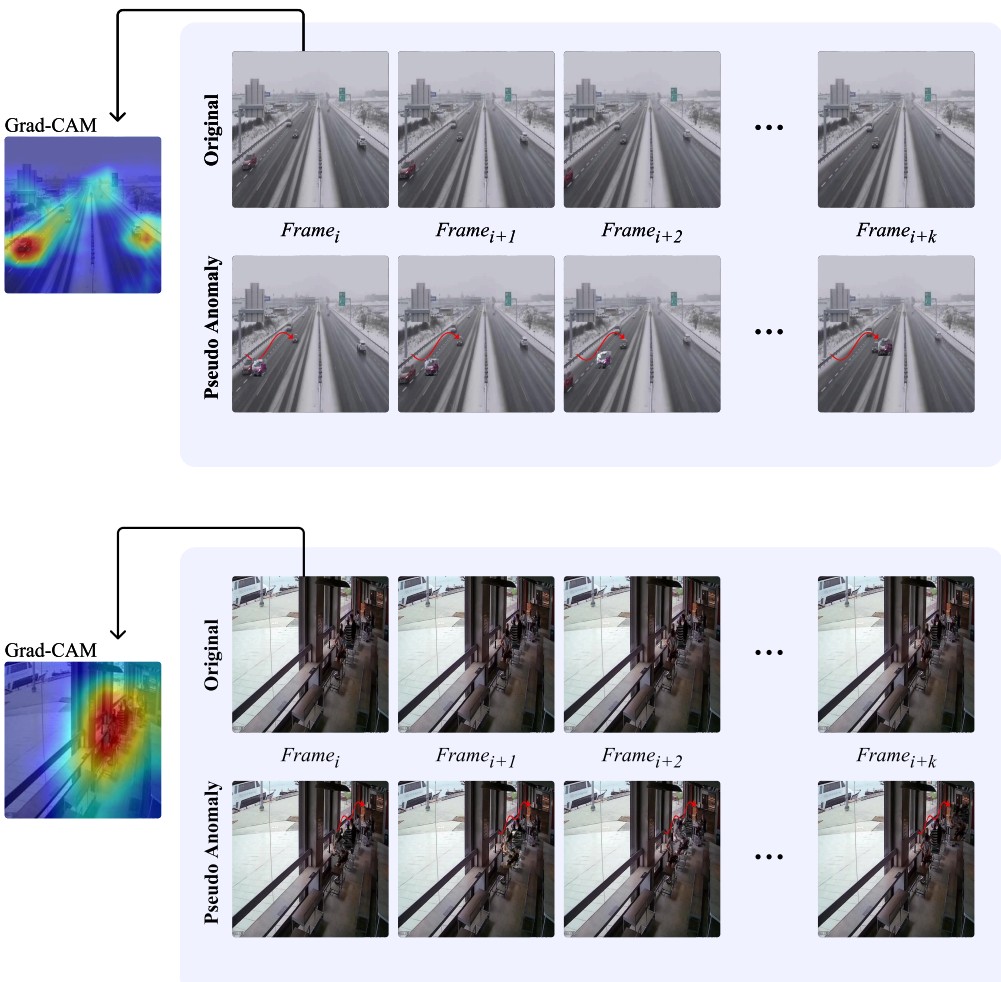

Figure 5: Visualization of pseudo-anomalous data generated by the SRD module. The top row shows the original (normal) sequence of frames along with the Grad-CAM heatmap of the first frame. The bottom row displays the corresponding distorted (pseudo-anomalous) frames created by applying the SRD distortion strategy.

