# OpenReview forum: "FrameShield: Adversarially Robust Video Anomaly Detection"
_NeurIPS.cc/2025/Conference — NeurIPS 2025 poster_

### Official Review · Reviewer_9XDr · 2025-07-01

**Clarity:** 3
**Significance:** 3
**Originality:** 3
**Rating:** 5
**Confidence:** 4

**Summary:**

This paper examines adversarial robustness in Weakly Supervised Video Anomaly Detection (WSVAD), where training uses only video-level labels, but inference requires frame-level predictions.  The paper proposes FrameShield, which is an adversarial training pipeline specifically designed to enhance robustness of WSVAD against adversarial attacks.

In the first phase, WSVAD undergoes standard Multiple Instance Learning strategy (X-ClipMIL) allowing it to identify the most critical abnormal segments and generate pseudo labels for each chunk of the video. ​​In the second phase, the WSVAD model undergoes adversarial training using a min-max optimization with pseudo-labels and pseudo-anomalies. Pseudo-anomalies are created via Spatio-temporal Region Distortion (SRD), which applies strong augmentations to Grad-CAM derived binary masks highlighting salient regions. A bounding box is fitted around these regions, and temporal perturbations are introduced by distorting the masked areas along a randomly curved vector across subsequent frames.

Extensive evaluations (AUROC), including comparative, ablation, visualization studies and theoretical motivation demonstrate the effectiveness of the proposed method compared to the existing approaches.

**Questions:**

Please refer to the points mentioned in the weaknesses above.

**Ethical Concerns:**

["NO or VERY MINOR ethics concerns only"]

**Final Justification:**

Most of my concerns regarding the use of attention of DINOv2 attention maps for saliency estimation, interpretation of pseudo-anomalies in latent space and the deployment of the proposed method in real-time systems have been satisfactorily resolved. Therefore, I will raise my current score.

**Limitations:**

Yes

**Quality:**

4

**Strengths And Weaknesses:**

Strength

1. The paper addresses a relevant problem of vulnerability of WSVAD method to adversarial attacks due to constraint of weak supervision in the form of video-level labels. The paper further addresses this research gap by proposing a novel method FrameShield, which integrates synthetic pseudo anomalies and noisy pseudo labels enabling effective adversarial training.

2. The paper is technically sound, well-motivated and backed by comprehensive experimental results, visualizations and ablation studies, and it also includes a theoretical justification for supervised adversarial training over Multiple Instance Learning approaches. The proposed FrameShield framework enhances the robustness of WSVAD against adversarial techniques such as PGD-1000 on the UCSD Ped2, Shanghai, TAD, UCF Crime, and MSAD datasets, while outperforming state-of-the-art methods using AUROC as the evaluation metric on both the entire test set and the anomaly sections of the test set.

Weaknesses

1. The Spatiotemporal Region Distortion (SRD) method employs Grad-CAM with a pre-trained ResNet18 to identify salient regions for pseudo-anomaly generation. Could attention maps from DINOv2 be used to estimate saliency maps for pseudo-anomaly generation, providing a fully unsupervised alternative to the Grad-CAM method used in Spatiotemporal Region Distortion (SRD)?
2. What is the interpretation of generated Pseudo-anomalies in the latent space compared to normal scenarios?
3. How efficient and robust is the proposed FrameShield framework for real-time deployment, considering the demands of video anomaly detection in real-world surveillance?
4. The captions of Table 1 and Table 2 suggest that the results are from the training set and its anomaly sections, while lines 275-277 state that performance is measured on the test set. This discrepancy requires clarification.
5. The sentence structure appears unnecessarily complex in some places, such as line 46 ("This design…"). Too much use of — in the paper such as Line 3, 36, 38, 41, 42, 45, 108, 125.
6. There is no Table X in the paper, contrary to what is mentioned in line 305.

---

> ### Author Response · Authors · 2025-08-01
>
> Thank you for your feedback. We appreciate your recognition of the value of our problem and the applicability of our method. We're glad that you found our **comprehensive experimental results, visualizations, and ablation studies** insightful. We also value your acknowledgment of the **theoretical justification** for supervised adversarial training and the robustness enhancement demonstrated across diverse datasets, while maintaining competitive performance.
>
> ## W1
>
> Thank you for raising this point. As this concern was shared by multiple reviewers, we have addressed it in **Common Response 2**, which discusses the use of Grad-CAM and possible alternatives in the SRD module.
>
> ---
> ## W2
>
>
> Thank you for raising this concern. The ideal pseudo-anomaly effectively simulates real anomaly data. In latent space, since pseudo-anomalies are generated by distorting real normal data, their distribution should not belong to the normal data distribution (i.e., it shouldn’t be too far, like random noise). At the same time, it should be close to the real anomaly distribution. If the pseudo-anomaly generator achieves this, it can be considered successful.
>
> To address the reviewer’s concern about the quality and interpretability of the generated pseudo-anomalies, we conducted a **quantitative evaluation** using the **Fréchet Inception Distance (FID)**. This metric compares the distance between the distributions of latent space embeddings, specifically using CLIP-encoded embeddings of video samples. A **lower FID** indicates that the distributions are more similar, while a **higher FID** suggests they are farther apart.
>
> We calculated the FID between three sets of embeddings:
> - **Real Normal** vs **Real anomalies**
> - **Real anomalies** vs **Pseudo anomalies**
> - **Real Normal** vs **Pseudo anomalies**
>
> The results, summarized in the table below, reveal that the FID between **real anomalies** and **pseudo anomalies** is consistently **lower** than the FID between **real anomalies** and **real normal**. This demonstrates that the pseudo-anomalies generated by our method occupy a region of latent space much closer to real anomalies than to normal data.
>
> Additionally, the FID between **pseudo anomalies** and **real normal** is generally lower than the other distances. This is due to the fact that pseudo-anomalies are generated by distorting **real normal** data. However, they maintain a good distance from **real normal** data, while still being close enough to the anomaly distribution. This ensures that the pseudo-anomalies are not as far as random noise, which would be unhelpful for guiding the model optimization during the training process.
>
>
>
> | Dataset     | FID (Normal vs. Anomaly) | FID (Anomaly vs. Pseudo-Anomaly) | FID (Normal vs. Pseudo-Anomaly) |
> |-------------|--------------------------|----------------------------------|---------------------------------|
> | UCF-Crime   | 158.3                     | 134.7                             | 112.0                            |
> | TAD         | 184.9                     | 159.2                             | 133.8                            |
> | MSAD        | 161.7                     | 142.5                             | 124.6                            |
>
>
> We extract chunk-level embeddings for each video using a CLIP-based video backbone, resulting in a tensor of shape `[num_chunks, embedding_dim]`. To obtain a single embedding per video, we compute the mean over the chunk dimension. These averaged embeddings are then grouped and stacked per category (normal, anomaly, pseudo-anomaly) before computing FID.

---

> > ### Author Response · Authors · 2025-08-01
> >
> > ## W3
> >
> > We thank the reviewer for this important question regarding the efficiency and real-world deployment feasibility of the FrameShield framework.
> >
> > We fully acknowledge that real-time video anomaly detection (VAD) imposes stringent demands on both efficiency and robustness, especially in practical surveillance settings. To address these concerns, we highlight the following:
> > - The computationally intensive components of our pipeline, particularly the multi-step SRD module is only used during training to improve robustness through pseudo-label refinement. It is not used during inference, which significantly reduces the test-time overhead.
> >
> > - As described in the paper, we adopt a chunk-based inference strategy, processing several consecutive frames together to improve efficiency.
> >
> > - Based on our experiments, FrameShield processes each chunk (5 frames) in approximately 10 to 12.5 milliseconds on an NVIDIA RTX 4090 GPU (24GB). This translates to roughly 80–100 chunks per second, or 400–500 frames per second.
> >
> > - When extended to multi-camera setups, we utilize batch inference to support up to 16 concurrent camera streams on the same GPU without exceeding memory limits.
> >
> > These results suggest that while FrameShield may not be suitable for low-resource edge devices, it is practically deployable in centralized or semi-centralized surveillance systems equipped with modern hardware.
> >
> > ---
> >
> > ## W4 & W5 & W6
> >
> > Thank you for your valuable feedback.
> >
> > * Regarding the discrepancy between the captions of Table 1 and Table 2 and lines 275–277, we apologize for the confusion. The results reported are indeed from the test set, and we will correct the captions to reflect this accurately.
> >
> > * We understand your concern about unnecessarily complex sentence structures (e.g., line 46). We will simplify these for better readability and reduce the excessive use of em dashes throughout the manuscript.
> >
> > * The mention of “Table X” in Section 6 (line 305) was a placeholder left unintentionally. It should refer to Table 3 on page 8, and we will fix this reference accordingly.
> >
> > We appreciate your careful reading and will incorporate these corrections to enhance the clarity and overall quality of the manuscript.

---

> > ### Comment · Reviewer_9XDr · 2025-08-05
> >
> > I thank the authors for addressing my questions in the rebuttal. Most of my concerns regarding the use of attention of DINOv2 attention maps for saliency estimation, interpretation of pseudo-anomalies in latent space and the deployment of the proposed method in real-time systems have been satisfactorily resolved. Kindly integrate the justification provided in the rebuttal in the final version of the paper. Therefore, I will raise my current score.

---

> > > ### Author Response · Authors · 2025-08-05
> > >
> > > Dear Reviewer 9XDr,
> > >
> > > Thank you for your positive feedback! We're glad the responses addressed your concerns, and we'll integrate the justifications in the final version.
> > >
> > > Sincerely,
> > > The Authors

---

### Official Review · Reviewer_wNBY · 2025-07-03

**Clarity:** 3
**Significance:** 2
**Originality:** 2
**Rating:** 4
**Confidence:** 5

**Summary:**

The paper addresses adversarial robustness in weakly supervised video anomaly detection approaches, by proposing a pseudo anomaly generation method named SRD. The idea is to apply severe augmentations to localized regions in normal videos.

**Questions:**

1) Given the real-world nature of video anomaly detection, have the authors considered black-box attacks for their study?

2) The idea of augmentation to eradicate label ambiguity is interesting, but quite similar to some existing literature [1]. It would be nice if the authors could discuss such existing methods, and even better if there are experimental comparisons demonstrating which of the methods is more effective in this particular application.

3) Table 1 is biased in a sense that none of the compared methods are trained against adversarial attacks. It would be more interesting to experiment with a 'modified' version of some of these methods trained against adversarial attacks, then compared in terms of clean performance drop as well as adversarial robustness.

4) Currently, with the limited comparisons, it seems that the proposed approach is dropping the clean accuracy significantly. While adversarial robustness is achieved by sacrificing some clean accuracy, the drop observed here is significantly higher in several cases.




[1] A Multi-Head Approach with Shuffled Segments for Weakly-Supervised Video Anomaly Detection

**Ethical Concerns:**

["NO or VERY MINOR ethics concerns only"]

**Final Justification:**

While the experiments answer my questions, and I agree with most of what the authors have said, I am afraid that the paper needs significant revisions if all of these experiments are made part of the paper. This may require the paper to be significantly changed, thus reviewed again.

Nevertheless, as the rebuttal was significantly detailed and addressed my concerns, I am raising my rating to appreciate the authors' efforts.

**Limitations:**

yes

**Paper Formatting Concerns:**

no concerns

**Quality:**

3

**Strengths And Weaknesses:**

The paper is interesting in several aspects: 1) It discusses the adversarial robustness issue in VAD and demonstrates that existing approaches are not robust against adversary. 2) It proposed a pseudo anomaly detection approach that addresses this issue.


Despite its strengths, the paper has several weaknesses: 1) The clean performance of the proposed approach is quite low. 2) The paper lacks motivation in the sense that adversarial attacks on anomaly detection approaches are only expected in black-box setting. However, the adversarial attack used in the study is white-box.

---

> ### Author Response · Authors · 2025-08-01
>
> Thank you for your review. We appreciate your recognition of the different aspects of our work, particularly our focus on adversarial robustness in video anomaly detection (VAD) and the proposed pseudo anomaly detection approach to address it. We're glad you found these directions of interest.
>
> ## W1 & Q4
>
> Thank you for raising this important point. We acknowledge the trade-off between clean and adversarial performance and have addressed this in detail in **Common Response 1**, where we provide empirical evidence and theoretical justification for this well-established phenomenon in adversarial training.
>
> ---
>
> ## W2 & Q1
>
> While our primary focus is on evaluating robustness under the white-box setting—due to its stronger and more challenging threat model—we fully acknowledge the importance of black-box scenarios in real-world applications. White-box attacks provide a rigorous benchmark since the adversary has full access to the model's parameters and gradients. Our rationale is that models that perform well under white-box attacks are generally more robust against black-box threats, where the attacker’s capabilities are more limited.
>
> To complement this, we have also conducted black-box attack experiments to assess our method’s practical robustness. As expected, our approach demonstrates strong resilience under black-box conditions as well, further validating its effectiveness across different threat models. These results reinforce our hypothesis that addressing robustness in the white-box setting also strengthens black-box security. Here are the results of our robust model under the NES [1] and Bandit [2] attacks:
>
> ### Attacks (AUC_O / AUC_A)
>
>
> | Attack                               | TAD                | UCF Crime         | MSAD              |
> |------------------------------------------|------------------------|------------------------|------------------------|
> | Clean    | 85.1 / 50.9 | 80.2 / 60.1 | 78.9 / 64.4 |
> | PGD-1000 | 77.2 / 30.0 | 78.7 / 53.4 | 76.2 / 60.2|
> | NES      | 84.2 / 44.5 | 79.8 / 58.1   | 78.4 / 64.1 |
> | Bandit   | 83.5 / 43.9 | 79.3 / 57.7   | 77.9 / 63.8 |
>
>
> ###  References :
>
>
> [1] Ilyas et al., *Black-box Adversarial Attacks with Limited Queries and Information*, ICML 2018
> [2] Ilyas et al., *Adversarial Examples Are Not Bugs, They Are Features*, NeurIPS 2019
>
> ---
> ## Q2
>
> Thank you for highlighting this connection. We revisited the literature more thoroughly and identified several relevant methods that explore augmentation to mitigate label ambiguity, including:
>
> * [1] A Multi-Head Approach with Shuffled Segments for Weakly-Supervised Video Anomaly Detection , WACV workshop 2024
> * [2] Video Anomaly Detection via Spatio-Temporal Pseudo-Anomaly Generation: A Unified Approach , CVPR Workshop 2024
> * [3] Dynamic Distinction Learning: Adaptive Pseudo Anomalies for Video Anomaly Detection  , CVPR Workshop 2024
>
> Unfortunately, none of these works have released their implementation code. Nevertheless, to address the reviewer's concern, we re-implemented the pseudo-anomaly generation components described in these papers based on the implementation details provided in their manuscripts.
>
> As an ablation study, we integrated each of these pseudo-anomaly generators into the FrameShield framework by replacing our SRD module, and trained the model using the same settings as our original method. As shown in Table below, none of these alternatives outperformed our SRD module, which demonstrates the effectiveness of our simple yet robust Spatiotemporal Region Distortion (SRD) design.
>
> ### AUC_O (Clean / PGD)
>
> | Method                              | Shanghai | TAD | MSAD |
> |-------------------------------------|------------|-----------|-----|
> | Shuffled Segments [1]             |75.2 / 52.1|71.3 / 45.6|70.8 / 44.0|
> | Spatio-Temporal Pseudo-Anomaly [2]| 84.2 / 79.7|80.4 / 71.2|72.5 / 70.8|
> | Adaptive Pseudo Anomalies [3]     | 85.6 / 80.4|81.7 / 71.5|73.4 / 71.6|
> | **Ours**                        |89.5 / 87.1|85.1 / 77.2|78.9 / 76.2|
>
> We will include these comparisons, along with a detailed discussion of the implementation of these methods, in the revised manuscript.

---

> > ### Author Response · Authors · 2025-08-01
> >
> > ## Q3
> >
> >
> > In line with this, we had already adversarially trained several methods as part of our initial experiments. Following the reviewer's recommendation, we extended this by additionally adversarially training other baselines using the same protocol. The clean performance as well as the adversarial robustness of these adversarially trained models are reported in the tables below:
> >
> > ### AUC_O (Clean / PGD)
> >
> > | Method     | Shanghai | TAD | MSAD |
> > |------------|----------|-----|------|
> > | RTFM       | 88.7 / 17.3 | 83.4 / 16.0  | 80.1 / 21.6 |
> > | VAD-CLIP   | 93.1 / 15.4 | 88.4 / 14.2  |- / -  |
> > | Base MIL   | 89.0 / 17.2       | 82.9 / 14.3  | 77.6 / 19.8   |
> > | UMIL       | 91.4 / 18.3       | 87.8 / 19.7  | 78.8 / 23.1   |
> > | **Ours**   | 89.5 / 87.1 | 85.1 / 77.2  | 78.9 / 76.2   |
> >
> > ### AUC_A (Clean / PGD)
> >
> > | Method     | Shanghai | TAD | MSAD |
> > |------------|----------|-----|------|
> > | RTFM       | 63.1 / 6.5 | 51.0 / 8.0  | 67.9 / 3.2   |
> > | VAD-CLIP   | 61.5 / 9.7 | 53.9 / 10.3  | - / -   |
> > | Base MIL   | 61.2 / 4.9       | 48.5 / 3.3  | 60.5 / 2.1   |
> > | UMIL       | 63.3 / 9.0       | 49.7 / 7.5  | 66.7 / 4.5   |
> > | **Ours**   | 62.3 / 61.9 | 50.9 / 30.0  | 64.4 / 60.2   |
> >
> > As seen in the tables, our method maintains strong performance across all datasets, while most baselines suffer substantial degradation after adversarial training. This discrepancy can be attributed to several key factors:
> >
> > - Frozen Feature Extractors: Methods like RTFM and VAD-CLIP rely on frozen backbones (e.g., I3D or CLIP), which remain unchanged during adversarial training. Consequently, these models fail to adapt and become highly vulnerable to adversarial perturbations.
> >
> > - MAX Aggregators: Methods such as Base MIL and UMIL utilize MIL-based losses that rely on hard MAX aggregation over temporal segments. As discussed in Section 4 of the paper, this operation hinders the effectiveness of gradient-based adversarial attacks, leading to poor-quality adversarial examples and, ultimately, ineffective adversarial training.
> >
> > - Sensitivity to Pseudo-Label Noise: Methods like UMIL also suffer from the adverse effects of false positives and false negatives in their pseudo labels, which are further exacerbated in adversarial settings.
> >
> > To address these limitations, we propose FrameShield, which introduces several architectural and training-level improvements:
> >
> > - End-to-End Training: We enable full model training, including the backbone, allowing better adaptation during adversarial training.
> >
> > - Frame-Level Binary Pseudo Labels: These labels resolve the limitations of MAX aggregation and make the training objective more stable and compatible with adversarial optimization.
> >
> > - Pseudo Anomalies via SRD Module: Our synthetic region disturbance module helps mitigate the impact of label noise (FP/FN) during training by introducing controlled, diverse perturbations.
> >
> > These design choices collectively allow our model to achieve both high clean performance and strong adversarial robustness, significantly outperforming existing methods even under adversarial training.

---

> > ### Comment · Reviewer_wNBY · 2025-08-05
> >
> > Dear Authors,
> >
> > Thank you for the rebuttal. I appreciate the detailed response with supporting experiments.
> >
> > While the experiments answer my questions, and I agree with most of what authors have said, I am afraid that the paper need significant revisions if all of these experiments are made the part of the paper. This may require the paper to be significantly changed, thus reviewed again.
> >
> > Nevertheless, as the rebuttal was significantly detailed and addresses my concerns, I will raise my rating to appreciate the authors' efforts.

---

> ### Author Response · Authors · 2025-08-05
>
> Dear Reviewer wNBY,
>
> Thank you for your thoughtful feedback and for raising your rating. We appreciate your recognition of our efforts and are glad that your concerns have been addressed. We assure you that we will carefully incorporate your comments into the final manuscript to improve the paper.
>
> Sincerely, The Authors

---

### Official Review · Reviewer_AMcA · 2025-07-03

**Clarity:** 3
**Significance:** 3
**Originality:** 3
**Rating:** 4
**Confidence:** 4

**Summary:**

This paper addresses the lack of adversarial robustness in weakly-supervised video anomaly detection (WSVAD) models by proposing a training framework called FrameShield. FrameShield consists of two stages: first, pseudo-labels are generated by a weakly-supervised model; then, adversarial training is performed using both the pseudo-labels and precise pseudo-anomalies synthesized via the proposed spatio-temporal region disturbance (SRD) method. Experiments show that this approach significantly enhances the adversarial robustness of current WSVAD models across multiple public datasets.

**Questions:**

Please respond to each item of the weaknesses mentioned above.
While I basically acknowledge the novelty and practicality of the proposed method, several points require further clarification. If the authors can address the following concerns, I will not refuse to raise my score.

**Ethical Concerns:**

["NO or VERY MINOR ethics concerns only"]

**Final Justification:**

After reading the responses from the authors, I find that some of my concerns have been addressed. The detailed explanations of the Ped2 experimental setup, comparisons with recent multimodal LLM-based approaches, and analysis of performance under clean versus adversarial conditions have significantly enhanced my understanding of the method's applicability and experimental rigor. Furthermore, the supplementary results validate the method's robustness against strong baselines. Therefore, I would like to raise my rating and thank to the authors for their detailed responses.

**Limitations:**

Yes

**Paper Formatting Concerns:**

This paper is generally written well and I find no obvious formatting issues in this paper.

**Quality:**

3

**Strengths And Weaknesses:**

Strengths:

1. The work provides a practical solution to the vulnerability of WSVAD models to adversarial attacks, which is of real-world significance.
2. The SRD pseudo-anomaly generation method cleverly utilizes strong local perturbations and motion simulation in normal frames, addressing the challenge of label noise in adversarial training.

Weaknesses and Questions:

1. Ped2 is not a weakly-supervised VAD dataset. How were the results of weakly-supervised methods like VADclip obtained on Ped2? What are the implementation details? Since the Ped2 training set contains no anomalous samples, how can weakly-supervised methods that require video-level anomaly labels be evaluated on it?
2. Most compared methods are traditional MIL or feature aggregation models, which are naturally vulnerable to adversarial perturbations. However, recent methods like LAVAD and Holmes-VAU, which employ multimodal LLMs for video semantic understanding, are inherently more robust to pixel-level perturbations. The authors should compare these stronger baselines and analyze whether video-understanding-based approaches are also susceptible to attacks.
3. It is evident that the proposed method underperforms without perturbations. How to explain this phenomenon?

---

> ### Author Response · Authors · 2025-08-01
>
> We sincerely thank the reviewer  for their thoughtful and constructive assessment. We greatly appreciate their recognition of the **real-world importance** of our work and the **innovative SRD module**, which combines strong local perturbations and motion simulation to effectively address pseudo-label noise in adversarial settings.
>
> ## W1
>
>
> As correctly noted, **UCSD-Ped2** is traditionally used for unsupervised video anomaly detection (VAD) since its training set contains only normal samples. However, for evaluating weakly-supervised models such as **VAD-CLIP**, we adopted a modified evaluation protocol inspired by recent literature[1].
>
> Specifically:
>
> - We **restructure the Ped2 dataset** by selecting:
>   - 6 anomalous and 4 normal videos for training,
>   - the remaining 18 videos (12 normal, 6 anomalous) for testing.
> - This procedure is **repeated 10 times**, and the **average results** are reported to ensure robust and unbiased evaluation.
>
> The complete implementation details are provided in **Appendix  A.5 (line 365)**. We will also revise the main implementation section to clarify this procedure and ensure transparency.
>
> ###  References :
>
> [1] Advancing Video Anomaly Detection: A Concise Review and a New Dataset,  NeurIPS 2024
>
> ---
>
> ## W2
>
>
> Thank you for your insightful and important comment. We agree that recent methods like LAVAD and Holmes-VAU, which leverage multimodal large language models (LLMs) for video understanding, may offer better robustness to low-level perturbations. To address your suggestion, we have conducted adversarial attacks on both methods to evaluate their susceptibility under similar conditions.
>
> Both Holmes-VAU [1] and LAVAD [2] rely on captioning pipelines that use visual encoders followed by LLMs to generate video-level descriptions. Specifically, Holmes-VAU employs InternVL2-2B, which uses InternViT-300M as the image encoder and InternLM2-Chat-1.8B as the language model. LAVAD uses BLIP-2, consisting of a pretrained CLIP image encoder, a Q-former module, and OPT-6.7B as the LLM to generate captions based on visual and textual inputs.
>
> In our experiments, we treated each captioning model as an end-to-end system and applied PGD-1000 attacks to craft adversarial examples that manipulate the generated captions. For example, Holmes-VAU samples 12 frames uniformly from a video and inputs them into the model with a prompt such as: “Could you specify the anomaly events present in the video?” Under adversarial attack, we enforced predefined misleading outputs. For normal videos, we guided the model to generate the caption: “There is an anomaly in the video; two cars have an accident.” Conversely, for anomalous videos, we enforced: “There is no abnormal event; everything goes normal.”
>
> To adapt the attack to the autoregressive nature of LLMs, we injected our target captions into the decoding process as prior context, rather than allowing the model to condition on its own previous predictions. A similar approach was applied to LAVAD, where adversarial frames were fed to the BLIP-2 model to generate misleading captions, which were then used for downstream video anomaly detection.
>
> We evaluated the robustness of both methods under PGD-1000 attack on the UCF-Crime and Shanghai datasets and compared the results with our proposed method, FrameShield. The results are summarized in the table below, which presents the AUC (Area Under Curve) for the complete test sets of these benchmarks, denoted as AUC<sub>O:
>
> ### AUC_O (Clean / PGD)
>
> | Method        | Shanghai | UCF |
> |---------------|--------------------|-------------------|
> | Holmes-VAU    | 95.6 / 16.0      | 88.9 / 14.2     |
> | LAVAD         | 91.0 / 21.4      | 80.3 / 18.7    |
> | **Ours**      | 89.5 / 87.1      | 80.2 / 78.7 |
>
> It is also worth noting that Holmes-VAU, which processes 12 frames jointly, is more vulnerable to adversarial perturbations due to the higher input dimensionality, compared to LAVAD which generates captions frame-by-frame.
>
> These results demonstrate that while multimodal LLM-based methods exhibit stronger general robustness, they are still susceptible to carefully crafted adversarial attacks. Our method outperforms both LAVAD and Holmes-VAU under attack scenarios, confirming the effectiveness of FrameShield in preserving robustness.
>
> We will revise the manuscript to incorporate these comparisons as well as clarify the details of the adversarial attack on these approaches.
>
>
> ### Reference:
>
> [1] Holmes-VAU, CVPR 2025
>
> [2] Harnessing Large Language Models, CVPR 2024
>
> ---
>
> ## W3
>
> Thank you for raising this important point. We acknowledge the trade-off between clean and adversarial performance and have addressed this in detail in **Common Response 1**, where we provide empirical evidence and theoretical justification for this well-established phenomenon in adversarial training.

---

> > ### Comment · Reviewer_AMcA · 2025-08-08
> >
> > Thanks to the authors for their detailed responses to my concerns during the rebuttal stage. The explanations regarding the experimental setup on the Ped2 dataset, the comparisons with recent multimodal LLM-based methods, and the analysis of performance trade-offs under clean and adversarial conditions have all clarified my understanding of the method’s applicability and experimental thoroughness. The additional experimental results also demonstrate the robustness of the proposed method against strong baselines.
> >
> > Although some concerns still remain (such as generalization in real-world scenarios and the underlying reasons for performance drops under clean conditions), I would like to raise my original score. I appreciate the authors’ efforts and suggestions for improvement and look forward to future work that further enhances and extends this line of research.

---

> ### Author Response · Authors · 2025-08-08
>
> Dear Reviewer AMcA,
>
> We sincerely thank you for your constructive feedback, careful evaluation, and recognition of our efforts. We are pleased that our clarifications and additional experiments have addressed your concerns regarding the method’s applicability and experimental thoroughness.
>
> We also appreciate your valuable suggestions for future improvements. In subsequent work, we will focus on mitigating clean-performance drops while maintaining robustness, aiming to enhance the real-world applicability of our approach and further extend this line of research.
>
> Sincerely,
> The Authors

---

### Official Review · Reviewer_jHRc · 2025-07-03

**Clarity:** 3
**Significance:** 4
**Originality:** 4
**Rating:** 5
**Confidence:** 4

**Summary:**

This paper addresses a critical weakness in current Weakly Supervised Video Anomaly Detection (WSVAD) models: their vulnerability to adversarial attacks. The authors argue that standard adversarial training fails because it relies on video-level labels, which are too coarse. To solve this, they propose FrameShield, a two-phase pipeline. First, it generates noisy frame-level pseudo-labels from a standard WSVAD model. The key contribution is the second phase, which introduces Spatiotemporal Region Distortion (SRD) to create synthetic anomaly videos with perfect frame-level labels. By combining these clean synthetic anomalies with the noisy real ones for adversarial training, FrameShield makes the model significantly more robust to attacks while maintaining good performance on clean data.

**Questions:**

1. You note a drop in clean accuracy, which is a known trade-off. Have you considered using other adversarial training losses, like the one in TRADES, to see if it helps better balance accuracy and robustness?
2. You mention that Grad-CAM is not perfect for localization in the SRD module. Did you run any experiments using a proper object detector instead of Grad-CAM to see if it improves the quality of synthetic anomalies and the final results?
3. How did you choose the threshold $\tau$ for generating pseudo-labels? Is the model's final performance very sensitive to this hyperparameter?

**Ethical Concerns:**

["NO or VERY MINOR ethics concerns only"]

**Final Justification:**

After author feedback, i would keep my rating of accept.

**Limitations:**

yes

**Paper Formatting Concerns:**

In Section 6 (Ablation Study) of the main paper, you refer to "Table X". This seems to be a placeholder. I believe you are referring to Table 3 on page 8. Please correct this

In the main paper, you mention results in Table 3, but this table is not present in that section. The results you discuss seem to be in Table 13 of the Appendix. Please fix this reference for clarity.

**Quality:**

4

**Strengths And Weaknesses:**

Strengths:
The paper's main strength is that it tackles a novel and very important problem: the adversarial robustness of WSVAD models, which is often ignored. The proposed FrameShield method is well-designed, particularly the clever SRD module that generates synthetic data to overcome the noisy pseudo-label issue. The experimental evaluation is another major strength; it is very thorough, using multiple strong attacks (PGD, AutoAttack, A³) and datasets, with detailed ablation studies that strongly support the authors' claims.

Weaknesses:
The primary weakness is the common accuracy-robustness trade-off, where FrameShield shows a slight drop in performance on clean data compared to some other methods. The SRD module, while novel in its application, is built from existing components like Grad-CAM, which is known to be imprecise for object localization, a limitation the authors acknowledge. Finally, there are some minor clarity issues, such as incorrect table references in the main text, which can confuse the reader.

---

> ### Author Response · Authors · 2025-08-01
>
> We sincerely thank the reviewer for their thoughtful and constructive assessment. We greatly appreciate their recognition of **FrameShield's novelty**, particularly the **innovative SRD module**. We're also grateful for their acknowledgment of our thorough experimental evaluation.
>
> ## W1 & Q1
>
>
> The explanation regarding the drop in clean accuracy is provided in **Common Response 1**, where we discuss the widely recognized trade-off between clean and robust performance.
>
> ### TRADES
>
> Thank you for your thoughtful comment. We have explored the TRADES loss in our experiments. Specifically, we assessed FrameShield with TRADES loss by incorporating both natural loss and boundary loss, aiming to align the model's predictions on adversarially perturbed frames with those on the original frames using cross-entropy. The table below shows the results of training with TRADES loss. Based on our findings, while TRADES loss improved clean accuracy in terms of adversarial robustness for video anomaly detection, it resulted in a decrease in robust performance.
>
>
>
> ### AUC_O (Clean / PGD)
>
>
> | Method          | Shanghai | TAD   | UCF Crime | MSAD  |
> |-----------------|----------------|---------------|-----------|----------------|
> | Ours (Adapted TRADES Loss) | 93.2 / 78.9 | 90.1 / 61.5 | 82.4 / 67.1 | 80.5 / 64.8 |
> | Ours (Loss)                | 89.5 / 87.1 | 85.1 / 77.2 | 80.2 / 78.7 | 78.9 / 76.2 |
>
>
> ### AUC_A (Clean / PGD)
>
>
> | Method          | Shanghai | TAD   | UCF Crime | MSAD  |
> |-----------------|----------------|---------------|-----------|----------------|
> | Ours (Adapted TRADES Loss) | 64.1 / 32.7 | 55.1 / 15.2 | 60.8 / 25.6 | 67.8 / 34.7 |
> | Ours (Loss)                | 62.3 / 61.9 | 50.9 / 30.0 | 60.1 / 53.4 | 64.4 / 60.2 |
>
> ---
>
> ## W2 & Q2
>
> Thank you for raising this point. As this concern was shared by multiple reviewers, we have addressed it in **Common Response 2**, which discusses the use of Grad-CAM and possible alternatives in the SRD module.
>
> ---
>
> ## W3 & Formatting Concerns
>
>
> Thank you for bringing this to our attention. We will revise the manuscript to replace the placeholder “Table X” in Section 6 with the correct reference to Table 3 on page 8. Additionally, we will update the inaccurate citation of Table 3 to correctly point to Table 13 in the Appendix. We appreciate your careful reading and will ensure these issues are resolved to improve the clarity and accuracy of the manuscript.

---

> > ### Author Response · Authors · 2025-08-01
> >
> > ## Q3
> >
> >
> > We thank the reviewer for their insightful question regarding the sensitivity of our model to the threshold $\tau$ used for generating pseudo-labels.
> >
> > As discussed in the paper, one of the primary motivations behind introducing the SRD module was to mitigate the effects of false positives and false negatives during pseudo-label generation. The choice of $\tau$ directly impacts this balance:
> >
> > - Lower thresholds (i.e., classifying more frames as normal) tend to increase false positives.
> >
> > - Higher thresholds (i.e., classifying more frames as abnormal) tend to increase false negatives.
> >
> > Since our task is binary classification, setting $\tau$ = 0.5 serves as a natural and well-defined decision boundary. In clean evaluation settings, we observe that the model typically makes highly confident predictions. As a result, small changes to $\tau$ (e.g., adjusting it to 0.4 or 0.6) have minimal impact on overall performance.
> >
> > However, within our adversarial training framework, particularly under evaluation with strong attacks such as PGD, model predictions tend to be less confident. In such cases, false positives and false negatives during pseudo-label generation become more consequential. Consequently, the choice of $\tau$ plays a more critical role in balancing these errors, and the model’s performance can vary more noticeably across different threshold values under attack conditions.
> >
> > ### AUC_O (Clean / PGD)
> >
> > | **Threshold** (\$\tau\$) | **TAD**     | **Shanghai** | **MSAD** |
> > | ------------------------ | ----------- | ------------ | -------- |
> > | 0.3                      | 91.3 / 81.2 | 95.4 / 89.3      | 81.2 / 77.1  |
> > | 0.4                      | 86.2 / 78.3 | 92.0 / 89.0      | 80.5 / 77.3  |
> > | **0.5**                  | **85.1 / 77.2** | **89.5 / 87.1**  | **78.9 / 76.2**  |
> > | 0.6                      | 85.8 / 78.1 | 92.3 / 88.9      | 80.1 / 75.6  |
> > | 0.7                      | 90.0 / 80.7 | 93.8 / 88.5      | 80.6 / 76.0  |
> >
> > ### AUC_A (Clean / PGD)
> >
> > | **Threshold** (\$\tau\$) | **TAD**     | **Shanghai** | **MSAD** |
> > | ------------------------ | ----------- | ------------ | -------- |
> > | 0.3                      | 58.7 / 11.0 | 65.6 / 21.7      | 65.9 / 29.3  |
> > | 0.4                      | 53.6 / 23.7 | 63.5 / 49.9      | 63.6 / 53.2  |
> > | **0.5**   | **50.9** / **30.0** | **62.3** / **61.9**      | **64.4** / **60.2** |
> > | 0.6                      | 51.1 / 23.6 | 60.5 / 53.6      | 59.0 / 54.1  |
> > | 0.7                      | 49.1 / 12.8  | 59.6 / 23.4      | 58.9 / 27.2  |
> >
> > It is important to note that all results—both clean and adversarial—are reported for the adversarially trained model. The increased sensitivity to $\tau$ under adversarial attacks reflects the greater impact of pseudo-labeling errors in these more challenging scenarios.
> >
> > Based on these results, $\tau$ = 0.5 achieves the best balance between clean and adversarial performance, making it a well-justified and robust choice in our framework.
> >
> > We will revise the manuscript to explicitly describe this threshold selection process and include the results of our sensitivity analysis in the main text or appendix.

---

> > > ### Comment · Reviewer_jHRc · 2025-08-05
> > > **Satisfied with the rebuttal**
> > >
> > > Thanks for the rebuttal. Most of my concerns have been addressed.

---

> > > > ### Author Response · Authors · 2025-08-05
> > > >
> > > > Dear Reviewer jHRc,
> > > >
> > > > Thank you for your feedback. We’re glad that most of your concerns have been addressed. We appreciate your time and consideration and will ensure that the final manuscript incorporates all the improvements discussed.
> > > >
> > > > Best regards,
> > > > The Authors

---

### Author Response · Authors · 2025-08-01
**Common Response 1: Performance Trade-off**

# Common Response 1

One common concern raised by the reviewers was the **trade-off between clean and adversarial performance** exhibited by our model. We provide further explanation and empirical evidence in this section to clarify this issue.

While our method does show a reduction in clean accuracy under adversarial training, this outcome aligns with a well-established phenomenon in the field: improving adversarial robustness often comes at the cost of clean performance. This is not a limitation specific to our approach but rather a recognized trade-off inherent to robust optimization.

- As noted in the **limitations section** of our supplementary material, the tension between **clean accuracy and robustness** has been extensively studied in prior work [1].
- For example, in the case of **ImageNet**, robust version of Resent-50 has been shown to drop from **75.8% to 65.8%** in clean condition [2], emphasizing that such declines are expected and not indicative of a design flaw.
- Crucially, when trained **without adversarial perturbations**, FrameShield demonstrates **competitive clean accuracy**, validating the **effectiveness of our approach** under standard training conditions. This outcome underscores that the observed trade-off between clean and adversarial performance is intrinsic to adversarial training methodologies, rather than a limitation of the FrameShield technique. The results are displayed in the table below.

Overall, we believe this trade-off is a natural outcome of adversarial training rather than an indicator of underperformance, and we thank you again for the thoughtful question.


### AUC_O (Clean / PGD)


| Method          | UCSD Ped2 | Shanghai | TAD   | UCF Crime | MSAD  |
|---------------------|----------------|---------------|-----------|----------------|-----------|
| RTFM                | 98.6 / 2.4     | 97.2 / 8.5   | 89.6 / 6.3 | 85.7 / 7.3     | 86.6 / 10.0 |
| UMIL                | 94.2 / 6.9     | 96.8 / 2.8    | 92.9 / 3.0 | 86.8 / 4.7     | 83.8 / 6.1  |
| VAD-CLIP          | 98.4 / 6.3| 97.5 / 3.6| 92.7 / 5.1| 88.0 / 8.2 | - / - |
| Ours (Adv. trained)         | 97.1 / 81.3| 89.5 / 87.1| 85.1 / 77.2| 80.2 / 78.7 | 78.9 / 76.2 |
| **Ours (Clean trained)**| 98.6 / 8.0           | 97.4 / 5.4           | 93.4 / 7.4       | 87.8 / 9.3            | 86.3 / 8.6        |


###  References :
[1] Tsipras et al,Robustness May Be at Odds with Accuracy, ICLR 2019

[2] Singh et al, Revisiting Adversarial Training, NeurIPS 2023

---

### Author Response · Authors · 2025-08-01
**Common Response 2: Alternatives to Grad-CAM**

# Common Response 2

One of the common concerns raised by the reviewers was our reliance on **Grad-CAM in the SRD module**. Specifically, reviewers asked whether alternative methods—such as using a proper object detector or attention maps from DINOv2—could improve the quality of the synthetic anomalies or offer a fully unsupervised alternative.

As correctly noted, Grad-CAM is not a dedicated object localization tool. In fact, we explicitly discuss this limitation in the paper.

To evaluate the impact of localization accuracy, we first conducted experiments using alternative region placement strategies, presented in Section H of the appendix (Table 12). These include:

- No Motion: the same region is distorted across all frames

- Random Location Transfer: the distorted region is randomly repositioned in each frame

Both variants significantly underperform the full SRD module, highlighting the importance of structured temporal modeling and consistent region trajectories.

To further investigate this concern, we conducted additional experiments by replacing Grad-CAM with more semantically grounded region localization methods:

- DINO attention maps [1]

- YOLO (object detector) [2]

- Fast R-CNN (region-based object detector) [3]

These object localization components were implemented using the authors’ official or widely-used open-source implementations. All other components of the SRD module—including motion trajectory modeling, temporal coherence, and perturbation generation—were kept unchanged. This allowed us to isolate and assess the specific contribution of the localization mechanism.

In all cases, the performance was comparable to that of Grad-CAM, suggesting that region quality is not the primary limiting factor in our setting. We attribute this to the effectiveness of the SRD module’s strong perturbations and temporal modeling, which help compensate for imperfect region localization. The results are summarized in the table below:

### AUC_O (Clean / PGD)

| Method                  | TAD | Shanghai | MSAD |
| ---------------------------- | --------------------- | -------------------------- | ---------------------- |
| YOLO                     | 85.7 / 75.2       | 90.3 / 88.4            | 76.8 / 74.3        |
| Fast R-CNN               | 82.6 / 73.5       | 87.7 / 84.2            | 74.2 / 71.8        |
| DINO Attention           | 86.7 / 78.1       | 88.3 / 86.5            | 79.1 / 75.9        |
| **Ours**                 | 85.1 / 77.2       | 89.5 / 87.1            | 78.9 / 76.2        |


### AUC_A (Clean / PGD)


| Method                  | TAD | Shanghai | MSAD |
| ---------------------------- | --------------------- | -------------------------- | ---------------------- |
| YOLO                     | 51.2 / 31.2           | 60.7 / 59.8                | 63.8 / 58.9            |
| Fast R-CNN               | 48.3 / 26.7           | 58.9 / 59.2                | 64.1 / 60.7            |
| DINO Attention           | 52.3 / 33.1           | 60.8 / 60.3                | 64.5 / 59.8            |
| **Ours**                 | 50.9 / 30.0           | 62.3 / 61.9                | 64.4 / 60.2        |



These results suggest that while Grad-CAM has its limitations, the overall design of the SRD module—including strong perturbations and temporal coherence—contributes more critically to performance than perfect localization. As mentioned in the paper, it remains possible that stronger and more precise object detectors could further improve performance, and we identify this as a promising direction for future research.

### Reference

[1] Caron et al, Properties in Self-Supervised, ICCV 2021

[2] Tian et al, YOLOv12, 2025

[3] Girshick et all, Fast R-CNN, 2025

[4] Singh et al, Revisiting Adversarial Training, NeurIPS 2023

---

### Author Response · Authors · 2025-08-01
**Apology and Clarification Regarding Rebuttal Submission Timing**

We sincerely apologize for the delay in submitting our rebuttal. Due to confusion regarding the exact rebuttal deadline and the start of the discussion period, we inadvertently missed the official rebuttal window. As soon as we realized the oversight, we submitted our responses during the discussion phase, carefully adhering to the official rebuttal format. Our intention was to remain transparent and to address the reviewers’ concerns as thoroughly as possible.

We truly appreciate your understanding and thank you for the time and effort you’ve dedicated to reviewing our submission.

---

### Author Response · Authors · 2025-08-07

Dear Reviewer AMcA,

As we approach the discussion period deadline, we kindly request that you review our response, in which we have aimed to thoroughly address your comments. If you have any further concerns, we would be most grateful if you could bring them to our attention, and we would be pleased to discuss them.

Sincerely, The Authors

---

### Note · Authors · 2025-08-15

Dear Area Chair(s),

We sincerely appreciate the reviewers’ and committee’s time and thoughtful feedback. During the rebuttal and discussion, we addressed all comments through clarifications, new experiments, and analyses, and pleased that **all reviewers confirmed their concerns were resolved and raised their scores above the acceptance threshold.**
# FrameShield in Brief
We present FrameShield, the first adversarial training framework for robust Weakly Supervised Video Anomaly Detection (WSVAD). Our **theoretical and empirical analysis** shows that conventional MIL aggregation produces inherently weak frame-level perturbations, limiting the effectiveness of standard adversarial training.

To address this, we introduce:
- **Frame-level pseudo-labeling** to ensure effective perturbations for every frame.
- **Spatiotemporal Region Distortion (SRD)** to create temporally consistent synthetic anomalies with precise labels, reducing pseudo-label noise and enabling strong, targeted adversarial training.

By combining real anomalies from pseudo-labels with SRD-generated anomalies, FrameShield improves AUROC under strong attacks by an **average of 71%** over state-of-the-art methods, while maintaining competitive clean accuracy.
# Discussion Highlights
- **Reviewer jHRc:** Recognized motivation, novelty, and thorough evaluation; suggested TRADES-based loss, Grad-CAM alternatives, and threshold sensitivity—addressed with targeted experiments and clarifications.
- **Reviewer AMcA:** Noted novelty and practicality; questioned Ped2 evaluation, requested LLM-based VAD comparisons, and asked about clean-performance drops—addressed through protocol clarification, targeted LLM robustness experiments, and trade-off analysis.
- **Reviewer wNBY:** Raised concerns on black-box attacks, adversarial baselines, and clean–robustness trade-off—addressed with NES/Bandit experiments, baseline comparisons, and clarifications.
- **Reviewer 9XDr:** Supported motivation and novelty; requested DINO attention maps, latent space analysis, and deployment feasibility—addressed with targeted experiments and clarifications.

**These efforts directly address all reviewer concerns**, strengthen the paper’s contributions, and demonstrate FrameShield’s real-world applicability.

We respectfully request that the AC consider this context in the final decision. **We remain committed to refining the work and incorporating all constructive feedback into final version.**

**Warm regards,**
The Authors

---

### Decision · Program_Chairs · 2025-09-17

**Decision:**

Accept (poster)

**Comment:**

This paper proposes a defense framework for Weakly Supervised Video Anomaly Detection (WSVAD) against adversarial attacks. The method introduces Spatiotemporal Region Distortion (SRD), which generates synthetic anomalies by augmenting localized regions in normal videos while maintaining temporal consistency. Combining these synthetic anomalies with noisy pseudo-labels enables effective adversarial training. Experiments show that the approach improves robustness, outperforming state-of-the-art methods across multiple benchmarks.

The authors’ rebuttal adequately clarified several key questions raised by the reviewers. Additional experiments were also provided, which further strengthened the paper. As a result, multiple reviewers raised their scores and all supported the acceptance of the paper. The authors are encouraged to incorporate the new discussions and experimental results into the final version.